# Aging measures and cancer in the Health and Retirement Study (HRS)

Shuo Wang[1,5], Anna Prizment [1,5] ✉, Puleng Moshele[1], Sithara Vivek [1], Weihua Guan[2], Anne H. Blaes [3], Heather H. Nelson[4] & Bharat Thyagarajan [1]

Cancer survivors may have higher biological age (BA) than cancer-free persons (controls). In HRS, we examined the associations of BA with cancer prevalence and mortality. BA was estimated by the Klemera and Doubal method (KDM-BA), phenotypic age (PhenoAge), and subjective age (SA) among 946 cancer survivors and 4555 controls; and by epigenetic clocks (Horvath, Hannum, Levine, GrimAge, Zhang Score (ZS), and methylation-based pace of aging (mPOA)) among 582 cancer survivors and 2805 controls. Age acceleration is estimated as residuals regressed on chronological age. There are significant multivariable associations with cancer prevalence for Hannum, GrimAge, and SA, and ZS (logistic regression), and with mortality for PhenoAge, Hannum, Levine, GrimAge, and ZS in cancer survivors, and for KDM-BA, PhenoAge, and ZS in controls (Cox regression). The strongest association in cancer survivors is for GrimAge (HR per 1 SD = 1.80, $p < 0.001$). PhenoAge and first- and second-generation epigenetic clocks hold promise for predicting mortality in cancer survivors.

Individuals diagnosed with cancer have started to live longer due to earlier cancer diagnoses, increased longevity, and improved cancer treatments; however, cancer survivors have more physiological dysfunctions compared to cancer-free individuals[1,2]. These dysfunctions may be caused by cancer treatment, the body's response to cancer (e.g., immunologic), the effects of cancer on the body (e.g., cachexia), the presence of unhealthy lifestyle factors, or an interaction between these factors[3–5]. Because individuals with the same chronological age (CA) may have very different physiological dysfunctions, the term biological age (BA) has been introduced. BA estimates the extent of accumulation of physiological damage in individuals with the same CA. BA has an advantage compared to CA because it is modifiable and may be reversed through anti-aging lifestyle interventions and medications that are currently under active investigation[6]. To estimate BA, multiple aging constructs have been developed including epigenetic clocks (ECs) based on DNA methylation profiles[7–11], proteomic aging clocks (using circulating proteins)[12–14], transcriptomic clocks (using gene

expression data)[15,16], and multidomain aging constructs comprised of clinical (biochemical, hematological, and physiological) markers[17,18]. Consistent with the hypothesis that cancer survivors have increased physiological dysfunctions, one previous study has found that ECs were higher among breast cancer survivors compared to similarly aged people without cancer[19]. To our knowledge, only a few studies have examined ECs and mortality in cancer survivors[20–22]. Additionally, different ECs were associated with incidence of cancer, including total cancer[10,23], colorectal[24], lung[18,24,25], breast[22,26], and pancreatic cancers[27], but the associations with cancer risk were inconsistent across different ECs and different study populations[10,24,26–29].

Besides ECs, multidomain aging metrics have been created: phenotypic age (PhenoAge) comprised of mortality-associated clinical biomarkers[18] as well as BA metric comprised of age-associated clinical biomarkers and computed by the Klemera and Doubal method (KDM-BA)[30]. These aging constructs, after adjusting for chronological age and other covariates, were associated with the risk of total, lung, and

[1]Department of Laboratory Medicine and Pathology, University of Minnesota, Minneapolis, MN, USA. [2]Division of Biostatistics and Health Data Science, School of Public Health, University of Minnesota, Minneapolis, MN, USA. [3]Division of Hematology, Oncology and Transplantation, Department of Medicine, University of Minnesota, Minneapolis, MN, USA. [4]Division of Epidemiology and Community Health, School of Public Health, University of Minnesota, Minneapolis, MN, USA. [5]These authors contributed equally: Shuo Wang, Anna Prizment. ✉e-mail: prizm001@umn.edu

colorectal cancers in the UK Biobank study, a large prospective cohort study[31]. Additionally, PhenoAge was associated with the risk of breast cancer in that study[31]. To our knowledge, no studies have evaluated the utility of KDM-BA among cancer survivors, while a study in NHANES found that PhenoAge was higher among cancer survivors of any type cancer than cancer-free participants and age acceleration for PhenoAge was associated with an increased mortality risk in cancer survivors[32]. However, that study did not compare PhenoAge across specific cancer types or cancer treatments.

In addition to biomarker-based aging measures, subjective age (SA) has been studied in relation to cancer. SA estimates how old people feel, i.e., an individual's self-perception of their own age[33,34]. SA was 2% higher than CA in people diagnosed with cancer and 8% lower than CA among cancer-free controls[35]. Moreover, higher SA has been correlated with worse quality of life in cancer survivors[34,35]. In our prior work, we have shown that higher SA is associated with adverse biomarker profiles[33], and other studies have reported that SA is associated with negative health outcomes, including higher mortality in the general population[36,37], but its contribution to mortality in cancer survivors has not been examined.

Therefore, although measures of BA, such as PhenoAge, KDM-BA, ECs, and SA, were validated, the number and the scope of studies examining aging metrics in cancer survivors are limited and those studies were not able to compare the performance of various BA measures in those with and without cancer within the same study. Hence, we compared nine aging constructs, including KDM-BA, PhenoAge, SA, and six ECs (Horvath, Hannum, Levine, GrimAge, Zhang Score, and Dunedin methylation-based pace of aging, called mPOA in this study) in cancer survivors and cancer-free individuals in the Health and Retirement Study (HRS), a large nationally representative population-based cohort. Additionally, the longitudinal data collected in HRS prior to and after cancer diagnosis provide opportunities to examine associations with cancer risk and associations with mortality after cancer diagnosis. Thus, we examined the associations of these nine aging measures with all-cause mortality over four years among cancer survivors and controls, i.e., cancer-free individuals. Finally, we conducted an exploratory analysis to evaluate whether any of these aging constructs were associated with cancer risk in participants followed for four years.

## Results

The nine aging metrics examined in these study participants (age range: 56-90 years) are described in Supplementary Data 1. We used survey weights to account for the complex survey design used in the HRS study[38] and to make the study population representative of the entire HRS cohort[39–42]. In our analyses, we used the weights specific for the 2016 Venous Blood Study (VBS) in Sample A and the weights specific for the DNA methylation sample in Sample B (Sample A and Sample B are described below).

### Characteristics of study participants

Sample A included 5501 participants (946 cancer survivors, i.e., participants who reported being diagnosed with cancer at any time before 2016, and 4555 controls, i.e., those who reported having no history of cancer in 2016) with data on KDM-BA, PhenoAge, and SA (Supplementary Fig. 1). Sample B included 3387 participants (582 cancer survivors and 2805 controls) with data on ECs (Supplementary Fig. 2). Table 1 and Table 2 show the distributions of participants' characteristics by cancer status. Compared to controls, cancer survivors were more likely to be chronologically older. Cancer survivors also tended to be Non-Hispanic White and former smokers as well as to have lower physical activity, lower grip strength (dominant hand), and a higher comorbidity index that was constructed as the sum of seven self-reported conditions: hypertension, lung disease, cardiac disorders, stroke, arthritis, diabetes, psychiatric problems (Tables 1 and 2).

Among these seven conditions, cancer survivors were more likely to have hypertension, cardiac disorders, stroke, arthritis, and diabetes compared to cancer-free participants (Tables 1 and 2). In Sample A, 17.58% of cancer survivors self-reported having chemotherapy, 35.23%, having surgery, and 16.58%, having radiotherapy (Table 1). In Sample B, 16.85% of cancer survivors reported having chemotherapy, 33.03%, having surgery, and 16.23%, having radiotherapy (Table 2). Further, because we excluded participants for several reasons (as described in the Methods section and Supplementary Figs. 1 and 2), we described the distributions of main demographic and lifestyle factors among participants who were included and not included in the study. We found that included participants were more likely to be non-Hispanic White and have a higher education level in both Sample A and Sample B (Supplementary Table 1). In addition, there was a smaller percentage of current smokers among participants included in Sample A (Supplementary Table 1). The difference between participants who were included and those who were not might be explained by the fact that we included only those who answered questions about SA and characteristics of interest. Despite those differences, the cancer prevalence in our study samples was similar to the prevalence in the entire HRS cohort and in Surveillance, Epidemiology, and End Results (SEER) in the whole study and across different age groups (Supplementary Table 2). Among those older than 65, cancer prevalence of 20.5% in Sample A and 19.8% in Sample B was similar to the 20.1% prevalence in Medicare in the US[43].

### Aging constructs under study and age acceleration

We tested correlations between CA and nine aging constructs under study (KDM-BA, PhenoAge, SA, and six ECs: Horvath, Hannum, Levine, GrimAge, Zhang Score, and mPOA). Seven aging constructs, including KDM-BA, PhenoAge, SA, and four ECs (Hannum, Horvath, Levine, and GrimAge) were correlated with CA ($r = 0.60–0.92$) (Supplementary Tables 3 and 4). To examine the effect of these seven aging constructs independent of CA, we estimated age acceleration (abbreviated as Accel) as residuals after regressing each aging construct on CA. There was no correlation between CA and age acceleration for any of these aging constructs (Supplementary Tables 3 and 4). We did not calculate age acceleration for mPOA because it reflects the pace of the aging process and it was not correlated with CA ($r = 0.03$), or Zhang Score, because it is a mortality risk score that was only weakly correlated with CA ($r = 0.31$) (Supplementary Table 4).

Age acceleration for each aging construct as well as Zhang Score and mPOA were higher in cancer survivors than in controls but the difference was not statistically significant for KDM-BA (Tables 1 and 2). Among cancer survivors, those who received chemotherapy (vs. those who did not) had, on average, higher age acceleration for PhenoAge (PhenoAgeAccel), SA (SA-Accel), Horvath (HorvathAccel), Levine (LevineAccel), and GrimAge (GrimAgeAccel); however, these differences did not reach statistical significance (Supplementary Table 5). In addition, the mean LevineAccel and GrimAgeAccel were higher among those who received radiation therapy compared to those who did not, while the mean HorvathAccel was higher among those who had surgery compared to those without surgery, but none of these differences reached statistical significance (Supplementary Table 5). Although these non-significant results may reflect the true absence of differences by treatment status, the similar trends across different treatments and different aging measures suggest that the absence of significant differences may be explained by a limited sample size in these subgroup analyses.

### Associations between aging constructs and cancer prevalence

A participant was considered having prevalent cancer if they reported being diagnosed with cancer at any time before 2016. In our main analysis, we conducted two models and both of them accounted for survey weights. Model 1 was adjusted for chronological age, sex, and

**Table 1 | Distribution[a] of participants' characteristics by cancer status in 2016 in Sample A[b]; HRS**

| Participants' Characteristics | Controls (N = 4555) | Cancer survivors (N = 946) | P-value[f] |
|---|---|---|---|
| Chronological age (CA), years (SD) | 67.43 (0.23) | 71.01 (0.44) | **<0.001** |
| KDM-BA-Accel, years (SD) | −0.28 (0.06) | −0.11 (0.15) | 0.37 |
| PhenoAgeAccel, years (SD) | −0.60 (0.15) | 0.21 (0.27) | **0.020** |
| SA-Accel, years (SD) | −0.27 (0.20) | 1.09 (0.34) | **<0.001** |
| Female, % | 54.11 | 53.38 | 0.751 |
| Race/ethnicity, % | | | |
| Non-Hispanic White | 82.58 | 87.88 | **0.009** |
| Non-Hispanic Black | 7.59 | 6.16 | |
| Hispanic | 4.70 | 2.48 | |
| Other | 5.13 | 3.48 | |
| Education, % | | | |
| Less than high school | 10.51 | 10.80 | 0.978 |
| High school | 29.19 | 29.14 | |
| Greater than high school | 60.30 | 60.06 | |
| Body mass index (BMI), kg/m² (SD) | 29.73 (0.11) | 29.90 (0.23) | 0.472 |
| Smoking status, % | | | |
| Current smoker | 10.24 | 8.18 | **<0.001** |
| Former smoker | 42.86 | 50.69 | |
| Never smoker | 46.90 | 41.13 | |
| Ever drinker, % | 64.93 | 57.68 | **<0.001** |
| Physical activity[c], score (SD) | 30.13 (0.51) | 26.34 (0.79) | **<0.001** |
| Grip strength[d] (dominant hand), kg (SD) | 32.34 (0.22) | 30.55 (0.44) | **<0.001** |
| Comorbidity index[e], score (SD) | 1.87 (0.03) | 2.25 (0.05) | **<0.001** |
| Health conditions included in comorbidity index, % | | | |
| Hypertension | 55.09 | 60.03 | **0.009** |
| Lung disease | 9.31 | 11.52 | 0.163 |
| Cardiac disorders | 21.06 | 29.55 | **<0.001** |
| Stroke | 4.29 | 6.51 | **0.020** |
| Arthritis | 58.73 | 71.22 | **<0.001** |
| Diabetes | 20.11 | 22.98 | 0.107 |
| Psychiatric problems | 18.47 | 22.85 | **0.004** |
| CMV seropositivity, % | 59.38 | 61.12 | 0.520 |
| Cancer treatment | | | |
| Chemotherapy, % | NA | 17.58 | NA |
| Surgery, % | NA | 35.23 | NA |
| Radiotherapy, % | NA | 16.58 | NA |

*SD* standard deviation, *KDM-BA* biological age metric estimated by the Klemera and Doubal method, *PhenoAge* phenotypic age, *SA* subjective age, *Accel* age acceleration, *BMI* body mass index, *CMV* cytomegalovirus.
[a]Results were accounted for survey weights.
[b]Sample A included participants who reported their SA and had biomarker measures used to calculate KDM-BA and PhenoAge.
[c]Physical activity was calculated as a weighted sum of scores based on self-reported light, moderate, and vigorous activities.
[d]Grip strength was measured for half of participants in 2014 and for other half in 2016.
[e]Comorbidity index was assessed as a number of coexisting conditions, including hypertension, lung disease, cardiac disorders, stroke, arthritis, diabetes, and psychiatric problems, ranged from 1 to 7.
[f]When comparing cancer survivors and controls, *P-values* were calculated from two-sided chi-square test for categorical variables and two-sided t-test for continuous variables.

**Table 2 | Distribution[a] of participants' characteristics by cancer status in 2016 in Sample B[b]; HRS**

| Characteristics | Controls (N = 2805) | Cancer survivors (N = 582) | P-value[f] |
|---|---|---|---|
| Chronological age (CA), years (SD) | 67.90 (0.27) | 71.67 (0.51) | **<0.001** |
| HannumAccel, years (SD) | −0.12 (0.11) | 1.00 (0.28) | **<0.001** |
| HorvathAccel, years (SD) | −0.04 (0.15) | 0.77 (0.37) | **0.047** |
| LevineAccel, years (SD) | −0.17 (0.20) | 0.87 (0.33) | **0.014** |
| GrimAgeAccel, years (SD) | −0.46 (0.12) | 0.72 (0.24) | **<0.001** |
| Zhang Score, units (SD) | −1.15 (0.01) | −0.96 (0.02) | **<0.001** |
| mPOA, years of physiological decline per one chronological year (SD) | 1.06 (0.002) | 1.08 (0.005) | **0.007** |
| Female, % | 54.35 | 52.31 | 0.462 |
| Race/ethnicity, % | | | |
| Non-Hispanic White | 77.50 | 83.34 | **<0.001** |
| Non-Hispanic Black | 10.12 | 10.35 | |
| Hispanic | 6.25 | 3.08 | |
| Other | 6.13 | 3.23 | |
| Education, % | | | |
| Less than high school | 13.57 | 11.86 | 0.644 |
| High school | 29.25 | 29.89 | |
| Greater than high school | 57.18 | 58.25 | |
| BMI, kg/m² (SD) | 29.97 (0.13) | 29.88 (0.32) | 0.763 |
| Smoking status, % | | | |
| Current smoker | 11.25 | 9.24 | **0.020** |
| Former smoker | 42.92 | 51.14 | |
| Never smoker | 45.83 | 39.62 | |
| Ever drinker, % | 61.75 | 53.96 | **0.009** |
| Physical activity, score (SD) | 28.88 (0.61) | 24.19 (1.01) | **<0.001** |
| Grip strength (dominant hand), kg (SD) | 31.95 (0.38) | 30.43 (0.46) | **0.001** |
| Comorbidity index[c], score (SD) | 1.98 (0.04) | 2.44 (0.08) | **<0.001** |
| Health conditions included in comorbidity index, % | | | |
| Hypertension | 57.64 | 67.47 | **<0.001** |
| Lung disease | 9.65 | 14.00 | **0.023** |
| Cardiac disorders | 22.28 | 34.13 | **<0.001** |
| Stroke | 5.47 | 7.74 | **0.046** |
| Arthritis | 60.03 | 70.71 | **<0.001** |
| Diabetes | 23.06 | 27.61 | **0.041** |
| Psychiatric problems | 19.66 | 21.94 | 0.283 |
| CMV seropositivity, % | 62.04 | 65.58 | 0.29 |
| Cancer treatment | | | |
| Chemotherapy, % | NA | 16.85 | NA |
| Surgery, % | NA | 33.03 | NA |
| Radiotherapy, % | NA | 16.23 | NA |

*SD* standard deviation, *Accel* age acceleration, *mPOA* Dunedin methylation-pace of aging, *BMI* body mass index, *CMV* cytomegalovirus.
[a]Results were accounted for survey weights.
[b]Sample B included participants who had data on ECs.
[c]Physical activity was calculated as a weighted sum of scores based on self-reported light, moderate, and vigorous activities.
[d]Grip strength was measured for half of participants in 2014 and for other half in 2016.
[e]Comorbidity index was assessed as a number of coexisting conditions, including hypertension, lung disease, cardiac disorders, stroke, arthritis, diabetes, and psychiatric problems, ranged from 1 to 7.
[f]When comparing cancer survivors and controls, *P-values* were calculated from two-sided chi-square test for categorical variables and two-sided *t*-test for continuous variables.

**Table 3 | Associations of aging constructs with total cancer prevalence in 2016; HRS**

| Sample Aᵃ | | | | |
| --- | --- | --- | --- | --- |
| Aging constructs | No. of cancer survivors | No. of controls | OR (95% CI) per 1 SD increase in aging construct, p-valueᵇ | |
| | | | Model 1ᶜ | Model 2ᶜ |
| KDM-BA-Accel (SD = 3.83 years) | 946 | 4555 | 1.04 (0.94, 1.15), p = 0.389 | 0.99 (0.89, 1.10), p = 0.781 |
| PhenoAgeAccel (SD = 7.12 years) | | | **1.13 (1.03, 1.23), p = 0.008** | 1.04 (0.95, 1.16), p = 0.379 |
| SA-Accel (SD = 10.36 years) | | | **1.16 (1.07, 1.25), p = 0.0008** | **1.11 (1.02, 1.20), p = 0.018** |
| Sample Bᵃ | | | | |
| Aging constructs | No. of cancer survivors | No. of controls | OR (95% CI) per 1 SD increase in aging construct, p-valueᵇ | |
| | | | Model 1ᶜ | Model 2ᶜ |
| HannumAccel (SD = 5.22 years) | 582 | 2805 | **1.24 (1.10, 1.41), p < 0.001** | **1.22 (1.09, 1.37), p = 0.001** |
| HorvathAccel (SD = 6.39 years) | | | 1.13 (0.98, 1.29), p = 0.079 | 1.10 (0.97, 1.27), p = 0.117 |
| LevineAccel (SD = 6.77 years) | | | 1.13 (1.00, 1.29), p = 0.050 | 1.11 (0.97, 1.27), p = 0.122 |
| GrimAgeAccel (SD = 4.65 years) | | | **1.26 (1.10, 1.44), p < 0.001** | **1.25 (1.07, 1.48), p = 0.005** |
| Zhang Score (SD = 0.45 units) | | | **1.35 (1.17, 1.56), p < 0.001** | **1.31 (1.12, 1.54), p = 0.001** |
| mPOA (SD = 0.09 years of physiological decline per one chronological year) | | | 1.14 (0.99, 1.31), p = 0.062 | 1.10 (0.95,1.27), p = 0.195 |

*KDM-BA* biological age metric estimated by the Klemera and Doubal method, *PhenoAge* phenotypic age, *SA* subjective age, *Accel* age acceleration, *mPOA* Dunedin methylation-pace of aging, *SD* standard deviation, *OR* odds ratio, *CI* confidence interval, *BMI* body mass index, *CMV* cytomegalovirus.
ᵃSample A included participants who reported their SA and had biomarker measures used to calculate KDM-BA and PhenoAge. Sample B included participants who had data on epigenetic clocks (ECs).
ᵇP-values were calculated from multivariable logistic regression.
ᶜModel 1 was adjusted for chronological age, sex, and race/ethnicity. Model 2 was additionally adjusted for BMI, smoking status, ever drinking, physical activity, comorbidity index, as well as CMV infection. In the analysis of ECs, both models were additionally adjusted for the percentage of monocyte, neutrophil, and lymphocyte. All analyses were accounted for survey weights.

race/ethnicity (and in the analysis of ECs, additionally adjusted for monocyte percentage, neutrophil percentage, and lymphocyte percentage). Model 2 (fully adjusted) was further adjusted for education, BMI, smoking status, ever drinking, physical activity, comorbidity index, and cytomegalovirus (CMV) infection. In the analysis of ECs, we adjusted for most prevalent immune cells, neutrophils, lymphocytes, and monocytes, because immune cells have been shown to be associated with age acceleration for ECs in a recent study by Zhang et al. (2024)[44].

In Model 2 (fully adjusted), SA-Accel [odds ratio (OR) (95% confidence interval (CI)) per 1 SD = 1.11 (1.02–1.20)], age acceleration for Hannum (HannumAccel) [1.22 (1.09–1.37)], and GrimAgeAccel [1.25 (1.07-1.48)] was significantly associated with cancer prevalence (Table 3). Additionally, Zhang Score was associated with cancer prevalence [1.31 (1.12–1.54)] (Table 3). No association was observed for KDM-BA-Accel, PhenoAgeAccel, HorvathAccel, LevineAccel, or mPOA with cancer prevalence in Model 2 (Table 3). The regression coefficients for all the covariates adjusted for in the analysis of associations between aging constructs and cancer prevalence can be found in Supplementary Data 2. Considering the most common individual cancer types (lung, colorectal, breast, prostate), PhenoAgeAccel, GrimAgeAccel, and mPOA were significantly associated with prevalent lung cancer (Supplementary Table 6). GrimAgeAccel showed the highest OR with prevalent lung cancer: OR (95% CI) per 1 SD = 2.94 (1.57–5.50) (Supplementary Table 6). Additionally, PhenoAgeAccel [1.24 (1.05–1.47)] and GrimAgeAccel [1.62 (1.10–2.37)] was associated with prevalent colorectal cancer (Supplementary Table 6). GrimAgeAccel was associated with prevalent breast cancer [1.27 (1.03–1.67)], while PhenoAgeAccel was negatively associated with prevalent prostate cancer [0.83 (0.70-0.98)] (Supplementary Table 6). SA-Accel, HannumAccel, HorvathAccel, LevineAccel, and Zhang Score were not associated with the prevalence of any of the most common cancers (Supplementary Table 6).

## Associations between aging constructs and all-cause mortality in cancer survivors and controls

By 2020, there were 371 deaths in Sample A (122 cancer survivors and 249 controls) and 316 deaths in Sample B (103 cancer survivors and 213

controls). In Model 2 (fully adjusted), PhenoAgeAccel and Zhang Score were associated with mortality in both cancer survivors and controls; the overall pattern was that higher hazard ratio (HR) estimates were observed in cancer survivors compared to controls (Tables 4 and 5). For example, for Zhang Score: HR (95% CI) per 1 SD = 1.62 (1.27–2.06) in cancer survivors and 1.45 (1.21–1.75) in controls (Table 5). GrimAgeAccel [1.80 (1.37–2.39)], LevineAccel [1.56 (1.23–1.98)], and HannumAccel [1.37 (1.14–1.63)] was significantly associated with mortality among cancer survivors only, while KDM-BA-Accel [1.25 (1.04–1.50)] was significantly associated with morality among controls only (Tables 4 and 5). In Model 2, SA-Accel, HorvathAccel, and mPOA were not associated with mortality in either cancer survivors or controls (Tables 4 and 5). The regression coefficients for all the covariates adjusted for in the analysis of associations between aging constructs and mortality in cancer survivors and controls can be found in Supplementary Data 3 and 4.

In addition, we performed several stratified analyses (exploratory analyses) to evaluate whether the effect of aging measures on mortality among cancer survivors was modified by various demographics (sex and race/ethnicity), cancer treatment types (chemotherapy, surgery, radiotherapy), or time since cancer diagnosis. In the stratified analyses, among cancer survivors, sex modified the association with mortality for HannumAccel (p-interaction = 0.005), GrimAgeAccel (p-interaction = 0.002), and Zhang score (p-interaction = 0.001) (Supplementary Table 7). Race/Ethnicity (among non-Hispanic White and non-Hispanic Black individuals) modified the association between PhenoAgeAccel and mortality (p-interaction = 0.018) (Supplementary Table 8). In the stratified analysis of race/ethnicity, we did not include Hispanic participants and participants from other racial/ethnic groups, due to a limited number of deaths in these groups. In the stratified analysis of cancer treatment, compared to cancer survivors who did not receive chemotherapy, the associations with mortality among those who received chemotherapy was significantly stronger for SA-Accel (p-interaction = 0.015) and appeared to be stronger for PhenoAgeAccel with a borderline significance (p-interaction = 0.060) (Supplementary Table 9). Similar trends were observed for LevinAccel, GrimAgeAccel, and Zhang Score but these p-values for interactions with chemotherapy did not reach statistical significance

**Table 4 | Associations between aging constructs (KDM-BA, PhenoAge, and SA) and mortality among cancer survivors and controls in Sample A[a]; HRS (2016–2020)**

| Cancer survivors (N = 946) | | | | |
|---|---|---|---|---|
| Aging constructs | No. of deaths | Total person-year | HR (95% CI) per 1 SD increase in aging construct, p-value[b] | |
| | | | Model 1[c] | Model 2[c] |
| KDM-BA-Accel (SD = 4.07 years) | 122 | 3775 | **1.18 (1.01, 1.39), p = 0.040** | 1.05 (0.85, 1.28), p = 0.625 |
| PhenoAgeAccel (SD = 7.22 years) | | | **1.47 (1.17, 1.88), p = 0.002** | **1.37 (1.05, 1.78), p = 0.019** |
| SA-Accel (SD = 10.29 years) | | | 0.98 (0.68, 1.41), p = 0.910 | 0.82 (0.57, 1.17), p = 0.258 |
| Controls (N = 4555) | | | | |
| Aging constructs | No. of deaths | Total person-year | HR (95% CI) per 1 SD increase in aging construct, p-value[b] | |
| | | | Model 1[c] | Model 2[c] |
| KDM-BA-Accel (SD = 3.78 years) | 249 | 18,861 | **1.37 (1.18, 1.60), p < 0.001** | **1.25 (1.04, 1.50), p = 0.021** |
| PhenoAgeAccel (SD = 7.09 years) | | | **1.42 (1.25, 1.60), p < 0.001** | **1.29 (1.11, 1.49), p = 0.001** |
| SA-Accel (SD = 10.37 years) | | | 1.23 (0.97, 1.55), p = 0.085 | 1.08 (0.89, 1.32), p = 0.443 |

*KDM-BA* biological age metric estimated by the Klemera and Doubal method, *PhenoAge* phenotypic age, *SA* subjective age, *Accel* age acceleration, *SD* standard deviation, *HR* hazard ratio, *CI* confidence interval, *BMI* body mass index, *CMV* cytomegalovirus.
[a]Sample A included participants who reported their SA and had biomarker measures used to calculate KDM-BA and PhenoAge.
[b]P-values were calculated from multivariable Cox proportional hazards regression.
[c]Model 1 was adjusted for chronological age, sex, and race/ethnicity. Model 2 was additionally adjusted for BMI, smoking status, ever drinking, physical activity, comorbidity index, as well as CMV infection. All analyses were accounted for survey weights.

**Table 5 | Associations between epigenetic clocks and mortality among cancer survivors and controls in Sample B[a]; HRS (2016–2020)**

| Cancer survivors (N = 582) | | | | |
|---|---|---|---|---|
| Aging constructs | No. of deaths | Total person-year | HR (95% CI) per 1 SD increase in aging construct, p-value[b] | |
| | | | Model 1[c] | Model 2[c] |
| HannumAccel (SD = 5.48 years) | 103 | 2270 | **1.42 (1.15, 1.76), p = 0.002** | **1.37 (1.14, 1.63), p < 0.001** |
| HorvathAccel (SD = 7.01 years) | | | **1.26 (0.98, 1.62), p = 0.071** | 1.21 (0.94, 1.57), p = 0.136 |
| LevineAccel (SD = 6.88 years) | | | **1.64 (1.33, 2.03), p < 0.001** | **1.56 (1.23, 1.98), p < 0.001** |
| GrimAgeAccel (SD = 4.68 years) | | | **2.08 (1.60, 2.70), p < 0.001** | **1.80 (1.37, 2.39), p < 0.001** |
| Zhang Score (SD = 0.46 units) | | | **1.72 (1.32, 2.23), p < 0.001** | **1.62 (1.27, 2.06), p < 0.001** |
| mPOA (SD = 0.09 years of physiological decline per one chronological year) | | | 1.24 (0.98, 1.58), p = 0.073 | 1.14 (0.85, 1.54), p = 0.381 |
| Controls (N = 2805) | | | | |
| Aging constructs | No. of deaths | Total person-year | HR (95% CI) per 1 SD increase in aging construct, p-value[b] | |
| | | | Model 1[c] | Model 2[c] |
| HannumAccel (SD = 5.14 years) | 213 | 11,439 | 1.17 (1.00, 1.37), p = 0.050 | 1.13 (0.96, 1.31), p = 0.112 |
| HorvathAccel (SD = 6.25 years) | | | 1.07 (0.87, 1.30), p = 0.551 | 1.06 (0.87, 1.30), p = 0.555 |
| LevineAccel (SD = 6.74 years) | | | 1.08 (0.91, 1.29), p = 0.376 | 1.04 (0.86, 1.23), p = 0.647 |
| GrimAgeAccel (SD = 4.63 years) | | | **1.54 (1.32, 1.79), p < 0.001** | 1.18 (0.97, 1.44), p = 0.092 |
| Zhang Score (SD = 0.43 units) | | | **1.64 (1.39, 1.93), p < 0.001** | **1.45 (1.21, 1.75), p < 0.001** |
| mPOA (SD = 0.09 years of physiological decline per one chronological year) | | | **1.37 (1.16, 1.61), p < 0.001** | 1.15 (0.96, 1.36), p = 0.133 |

*Accel* age acceleration, *mPOA* Dunedin methylation-pace of aging, *SD* standard deviation, *HR* hazard ratio, *CI* confidence interval, *BMI* body mass index, *CMV* cytomegalovirus.
[a]Sample B included participants who had data on ECs.
[b]P-values were calculated from multivariable Cox proportional hazards regression.
[c]Model 1 was adjusted for chronological age, sex, race/ethnicity, as well as the percentage of monocyte, neutrophil, and lymphocyte. Model 2 was additionally adjusted for BMI, smoking status, ever drinking, physical activity, comorbidity index, as well as CMV infection. All analyses were accounted for survey weights.

(Supplementary Table 9); however, the number of deaths among cancer survivors who self-reported information about chemotherapy was limited. Although there were no statistically significant interactions of aging constructs with surgery (Yes/No) or radiation therapy (Yes/No) in relation to mortality among cancer survivors (Supplementary Tables 10 and 11), there was a pattern of stronger associations with mortality observed among cancer survivors who had surgery (or radiation) vs. those who did not have surgery (or radiation) for several aging constructs. Finally, time since cancer diagnosis (<2 years, 2-5 years, >5 years) did not modify the associations between any of the aging constructs and mortality in cancer survivors as the 95% CIs across time since cancer diagnosis largely overlapped for each aging construct (Supplementary Table 12).

To ensure that active cancer treatment at the time of blood draw was not a contributor to our findings for the associations of aging constructs with cancer prevalence and mortality among cancer survivors, we conducted a sensitivity analysis. In this sensitivity analysis, we excluded cancer survivors developed cancer within two years of blood collection – the time window where most cancer patients complete their treatment, and reran the analyses for cancer prevalence and

mortality in cancer survivors. We found that the results were comparable to the results in our main analyses for both cancer prevalence and mortality in cancer survivors (Supplementary Tables 13 and 14).

### Associations between aging constructs and cancer incidence

New cancer diagnoses were self-reported by 182 participants among controls in Sample A and 122 participants among controls in Sample B. None of the nine aging constructs was associated with incident cancer risk (Supplementary Table 15).

## Discussion

The number of individuals aged over 60 years is rapidly increasing in the U.S., so is the number of cancer survivors[45,46]. The number of cancer survivors aged 60 years or older increased from 8.4 million in 2008 to 14.1 million in 2022[47,48]. The aging process is characterized by a decline in physiological functions and the accumulation of age-related diseases. About 50% of individuals aged 65 years or older have two or more major age-related diseases, such as diabetes, cardiovascular disease, cancer, and/or macular degeneration, while 25% of them have three or more major age-related diseases, and these numbers are higher for those older than 75 years[49,50]. Cancer survivors experience even higher risk of morbidity and mortality compared to cancer-free individuals[4,5,51,52]. The declining health of cancer survivors is an unprecedented challenge to the health care system (see reviews[53,54] for a detailed discussion). Thus, there is a need in aging constructs that can measure biological age (BA) including those in cancer survivors. BA may be modifiable by anti-aging lifestyle interventions or drugs, such as senolytics and senomorphics, that are currently under active development[6]. Importantly, because aging process and age-related diseases share the same basic molecular mechanisms, intervening on BA is expected to slow down the development of multiple age-related diseases simultaneously[14]. This underscores the importance of identifying the optimal BA measure in older populations and particularly in cancer survivors.

In a large nationally representative cohort – the Health and Retirement Study (HRS), we examined the association of nine well-validated aging constructs with the prevalence of any cancer type and the most common cancers and with all-cause mortality in cancer survivors and individuals without cancer, i.e., controls. These constructs include two clinical-markers based constructs -- BA estimated by the Klemera and Doubal method (KDM-BA) and phenotypic age (PhenoAge), subjective age (SA), as well as six epigenetic clocks (ECs). These ECs include two first-generation ECs – Horvath and Hannum, three second-generation ECs – Levine, GrimAge, and Zhang Score, and one third-generation EC – methylation-based pace of aging (mPOA). Thus, our study provided a comparative analysis across multiple aging constructs, as well as across cancer survivors and cancer-free individuals. Additionally, we compared each aging construct and its association with mortality in cancer survivors across different cancer treatments. Furthermore, in our analysis of ECs, we adjusted for the most common immune cells (lymphocytes, neutrophils, and monocytes) that were measured at the same time as ECs. Immune cells were associated with age acceleration for ECs in a recent publication[44]; however, adjustment for immune cells in our study did not markedly change the associations.

In the cross-sectional analysis of aging measures in cancer survivors and cancer-free individuals (i.e., controls), we found that cancer survivors had significantly higher age acceleration (Accel) in eight of the nine aging constructs (except for KDM-BA). However, after multivariable adjustments, only four aging measures, SA-Accel, HannumAccel, GrimAgeAccel, and Zhang Score, were associated with cancer prevalence. Importantly, not only the second-generation ECs (GrimAge and Zhang Score), but also the first-generation EC (Hannum) and SA were associated with cancer prevalence. In our exploratory analysis of the most common individual cancers, PhenoAgeAccel,

GrimAgeAccel, and mPOA were significantly associated with prevalent lung cancer. Two of them (PhenoAgeAccel and GrimAgeAccel) were also significantly associated with prevalent colorectal cancer, and GrimAgeAccel was significantly associated with prevalent breast cancer. Among all aging measures, GrimAgeAccel showed the strongest associations with prevalent lung, colorectal and breast cancers.

In the longitudinal analysis of all-cause mortality in cancer survivors over four years of follow-up, higher PhenoAgeAccel, HannumAccel, LevineAccel, GrimAgeAccel, and Zhang Score were associated with higher all-cause mortality, with the strongest association for GrimAgeAccel. Our findings for PhenoAgeAccel were similar to the findings from a recent NHANES study that reported an increased risk of all-cause mortality in cancer survivors with higher PhenoAgeAccel[32]. For the fourth vs. the first quartile of PhenoAgeAccel, HR was 3.04 (95%CI: 2.31–4.00) in NHANES[32] and was 2.81 (95% CI: 1.31–6.06) in our study. In addition, the Melbourne Collaborative Cohort (MCC) study found that each 5-year increment in HannumAccel was associated with a 4–6% increased risk of all-cause death in cancer survivors[21]; the HR for HannumAccel in that study was lower than the estimate (HR per 5-year = 1.34) in our study. Furthermore, our findings suggest that both the first-generation (Hannum) and second-generation (Levine, GrimAge, Zhang Score) ECs are consistently associated with mortality in cancer survivors. However, in our study, second-generation ECs (i.e., GrimAge, Levine, and Zhang Score) showed stronger associations with mortality in cancer survivors compared to first-generation ECs (Hannum and Horvath). This may be explained by the fact that second-generation ECs were trained on mortality, while first-generation ECs were trained on chronological age. Thus, it is expected that second-generation ECs are more strongly associated with mortality[42].

Cancer survivors may experience accelerated aging due to both cancer itself and cytotoxic cancer treatments, particularly chemotherapy and radiation therapy, which may cause epigenetic changes and cellular senescence features such as telomere shortening and alterations in DNA repair genes[4,5]. In turn, these changes individually or in combination contribute to accelerated aging phenotypes in cancer survivors. In agreement with the hypothesized mechanisms, in our study, age acceleration for several aging constructs (PhenoAge, SA, Horvath, Levine and GrimAge) tended to be higher in cancer survivors who received treatment (chemotherapy, radiation, or surgery) compared to those who did not; however, all these differences did not reach statistical significance. In addition, in the analysis of mortality stratified by cancer treatment, we found a pattern of stronger associations with mortality for several aging constructs among cancer survivors with treatment vs. those without. Although not all results were significant, the similar trends observed for different treatments and different aging measures suggest that the absence of significant differences may be explained by a limited sample size in these subgroup analyses. An additional limitation of our analysis stratified by treatment is that cancer treatment was self-reported. Future studies with more accurate cancer treatment data are warranted to compare the aging process among cancer survivors with and without treatment, and thus could help improve cancer treatment protocols, which are important at both clinical and public health levels.

In our study, we examined a non-biomarker-based aging measure – SA. Age acceleration of SA (SA-Accel) was higher in cancer survivors (mean: 1.09 years) than controls (mean: −0.27 years) and SA-Accel was significantly associated with cancer prevalence in a model adjusted for demographics, lifestyle factors, physical activity, and medical conditions. The higher SA-Accel in cancer survivors may be contributed to by factors such as the stress of managing cancer, worries about cancer recurrence, life changes, job and family issues, or other factors. In our study, SA was correlated with CA and with other biomarker-based aging constructs, but these correlations were weaker than the correlations between the biomarker-based constructs. In our study, we did not find any associations between SA-Accel and mortality in cancer

survivors. To our knowledge, there were no studies examining the association between SA and mortality in cancer survivors. It is likely that SA captures different aspects of aging than biomarker-based aging constructs and may not accurately predict mortality in cancer survivors because they have a milieu of different age- and cancer-related issues that may not be captured by a one-time question about how old one feels.

Among cancer-free individuals in our study, KDM-BA-Accel, PhenoAgeAccel, and Zhang Score were associated with all-cause mortality. Consistent with our findings, higher KDM-BA-Accel and PhenoAgeAccel were associated with higher mortality in the NHANES study[55,56]. Compared to cancer survivors, in cancer-free individuals, there was a trend of slightly weaker associations with mortality for several ECs (Hannum, Levine, and GrimAge). For instance, in cancer survivors but not in cancer-free individuals, there were significant associations with mortality for HannumAccel, LevineAccel, and GrimAgeAccel. These findings support the conclusion from a recent NHANES study, which found a pattern of a stronger association between PhenoAgeAccel and mortality in cancer survivors than controls[32]. Most likely, specific features in cancer survivors, such as inflammatory processes associated with cancer itself and cancer treatments, may contribute to the aging process in cancer survivors. Moreover, in our study, only two aging measures (PhenoAge and Zhang Score) were associated with mortality in both cancer survivors and controls. Although the associations with mortality for PhenoAgeAccel in cancer survivors and controls were slightly weaker compared to those for Zhang Score, the relative inexpensiveness of PhenoAge measurement and easier implementation make it a promising option for future clinical applications. In summary, we found that different aging measures show different associations with mortality in cancer survivors and controls, suggesting that the appropriate aging measure to use may depend on the specific research questions being asked.

Aging and cancer may have a bidirectional association. Not only do cancer survivors have higher BA, but persons with higher BA may be at increased cancer risk, most likely because both cancer and aging have the same hallmarks -- genetic and epigenetic damage accumulation[57], inflammation, oxidative stress, and other damage[58–61]. However, contrary to the consistent associations of various aging constructs with prevalent cancer and mortality in our study, we did not find associations with total cancer incidence for any aging constructs. These findings do not agree with a previous study that found associations of different ECs, such as Horvath [HR per 1 year = 1.01, 95% CI (1.00–1.03)], GrimAge [1.07 (1.05–1.08)] and Levine [1.02 (1.01–1.03)], with the incidence of any type of cancer[62]. It is likely that the limited sample size in our study precluded us from detecting associations of this magnitude. Similarly, the limited sample size did not allow us to study the risk of individual cancers, such as lung or colorectal, for which the associations were stronger, as summarized in a recent review[63].

As already discussed, the most important strength of our study is the comparison of multiple aging measures in cancer survivors and cancer-free individuals within the same study. Importantly, our study examined aging measures in older cancer survivors, which represent a quickly growing group that needs cancer care tailored for this population. Our study was conducted within a population-based longitudinal study, i.e., the participants were followed prospectively, and aging measures were collected after cancer diagnosis and treatment. This allowed us to examine the risk of outcomes after a cancer diagnosis. Other strengths include detailed information about various risk factors, comorbidities, and cancer treatment, as well as multiple clinical and epigenetic biomarkers and immune cells measured at the same time. The limitation of this study included self-reported information about physician-diagnosed cancer and cancer characteristics. However, as mentioned in the results, the cancer prevalence in HRS

was similar to the cancer prevalence in SEER and Medicare. Moreover, recent validation studies in HRS found that, compared to Medicare data, self-report of first incident cancer diagnoses in HRS had 73% sensitivity and 99% specificity, suggesting that self-report of cancer data can be used for studying cancer in HRS[64,65]. Another limitation is that we have a limited sample size for the analysis of cancer risk and for the subgroup analyses of mortality in cancer survivors, especially for ECs that have been measured in a smaller sample.

In brief, our results suggest that different aging constructs may capture different aspects of aging and that cancer survivors and controls may experience different physiological dysfunctions. Notably, several aging constructs were more strongly associated with mortality in cancer survivors than controls. We conclude that PhenoAge (clinical-marker-based aging measure) and ECs from different generations (Hannum, Levine, Zhang score, and GrimAge) hold promise as potential measures of biological age in cancer survivors.

## Methods

### Inclusion & Ethics
This study complies with all relevant ethical regulations. All participants gave informed consent, and this study has been approved by the University of Minnesota Institutional Review Board.

### Study population
The HRS study is an ongoing biennial longitudinal study of a nationally representative sample of individuals over 50 years old in the United States, starting from 1992. At each survey, participants had either a face-to-face interview or an interview over the phone to report their demographics, lifestyle, and medical history. Subjective age (SA) data were obtained by responding to the question "How old do you feel?" for half of all participants in 2014 and for the other half in 2016. The biomarker data used to compute biological age estimated using the Klemera and Doubal method (KDM-BA) and phenotypic age (PhenoAge) were measured in blood samples collected from participants who completed the 2016 survey and attended the HRS 2016 Venous Blood Study (VBS) ($N = 9193$) (Supplementary Fig. 1). DNA methylation (DNAm) profiles were measured in a nonrandom subsample of participants ($N = 4018$) who attended the HRS 2016 VBS (Supplementary Fig. 2)[41]. A detailed description of the blood sample collection and processing is available on the HRS study website[40]. We used survey weights in all our analyses to account for the complex survey design used in the HRS study[38].

### Biological age constructs
**KDM-BA.** KDM-BA was developed by Levine et al. [2013][30] using 10 clinical markers that represent the decline in age-related physiological functioning and susceptibility to disease in old age (Supplementary Data 1). These 10 markers include systolic blood pressure (SBP), total cholesterol, fasting glucose, cytomegalovirus (CMV) infection, C-Reactive Protein (CRP), serum creatinine, blood urea nitrogen (BUN), alkaline phosphatase, albumin, and peak flow measurement. In this study, we used these 10 established biomarkers to compute KDM-BA and recalibrated the weights for the HRS sample using R package "BioAge"[66].

**PhenoAge.** PhenoAge was computed as a weighted sum of nine biomarkers (albumin, creatinine, glucose, log-transformed CRP (log CRP), lymphocyte percent, mean cell volume, red blood cell distribution width, alkaline phosphatase, and white blood cell count) as well as a person's CA (Supplementary Data 1)[56,67].

**Epigenetic Clocks (ECs).** DNAm in HRS was measured from DNA extracted from the buffy coat using the Infinium Methylation EPIC BeadChip at the University of Minnesota Genomics Center. The DNAm samples were randomized across analytic plates by age, cohort, sex,

education, and race/ethnicity, along with 39 pairs of blinded duplicates. Analysis of these duplicate samples showed a correlation > 0.97 across all CpG sites. Data preprocessing and quality of control were performed using the minfi package in R[41]. A total of 3.4% of the methylation probes were removed from the final data because their detection P-value fell below the threshold of 0.01 ($n = 29,431$ out of 866,091). The same DNAm measurements were used for all the individuals, and high-quality data were available for 4,018 individuals[41].

Thirteen published ECs were calculated in HRS using R scripts[41,68]. Among those 13 ECs, we selected six ECs that have been used in cancer studies: two first-generation ECs (Horvath[7] and Hannum)[8], three second-generation ECs (Levine[9], GrimAge[10], and Zhang score)[69], and a third-generation EC -- methylation-based pace of aging (hereafter called mPOA, also called DunedinPoAm)[11]. The original methods for the creation of these ECs are discussed in Supplementary Data 1. Horvath, Hannum, Levine, and GrimAge were expressed in years while mPOA was measured in "years of physiological decline occurring per 12 months of calendar time"[11]. Zhang Score is a mortality risk score that was not transformed to years.

### Ascertainment of outcomes: cancer and mortality

Cancer survivors were defined as participants who answered "yes" to the question "Has a doctor ever told you that you have cancer or a malignant tumor, excluding minor skin cancer?" on the 2016 survey. Participants who answered "no" to the above question were considered as controls, i.e., participants without cancer. Recent validation studies in HRS found that, compared to Medicare data, self-report of first incident cancer diagnoses in HRS had 73% sensitivity and 99% specificity, suggesting that self-report of cancer data can be used for studying cancer in HRS[64,65]. The month and year of cancer diagnosis, cancer site, and treatments were reported in all surveys conducted from 1992 to 2016[70]. Information about cancer treatments was collected in response to the question "Since [previous interview], what sort of treatments have you received for cancer?" Participant responses were categorized by survey technicians into eight categories: chemotherapy, surgery, radiation, medications/treatments for symptoms, biopsy, x-ray, other, or none. For our analysis, three indicator variables were created for chemotherapy, surgery, and radiation therapy. Participants who had any response to this question, but not chemotherapy (or radiation or surgery), were considered "no chemotherapy (or no radiation or no surgery)." Thus, the year and month of cancer diagnosis, cancer site, and cancer treatment (chemotherapy, surgery, and radiation) were collected from all surveys (1992-2016) and compiled for all eligible participants who reported being diagnosed with cancer.

In the prospective analysis of cancer risk, we included only participants who were cancer-free (i.e., controls) in 2016. We used the 2018 and 2020 surveys to identify incident cancer cases up to 2020. Participants who reported a new cancer diagnosis in either the 2018 or 2020 surveys were asked to report the month and year of their most recent cancer diagnosis. The earlier date reported in either survey was considered as the date of cancer diagnosis. For participants who died, the month and year of death were derived from interviews provided by household members.

### Other characteristics of interest

Race/ethnicity (non-Hispanic White, non-Hispanic Black, Hispanic, or Other), and education (less than high school (years at school <12 years); high school (12 years at school); greater than high school (years at school >12 years) were obtained from the Cross-Wave Tracker file. The following characteristics were obtained from the 2016 survey: CA (in years), sex (female/male), smoking status (current, former, or never smokers), and ever drinking (ever or never drinkers). Physical activity for each participant was calculated as a weighted sum of scores based on vigorous, moderate, and light physical activity, following the

methodology applied in a previous HRS study[71]. A comorbidity index was constructed using seven self-reported conditions diagnosed by a physician. These conditions included hypertension, lung disease, cardiac disorders, stroke, arthritis, diabetes, and psychiatric problems (yes/no). The measure of CMV seroprevalence was described previously[40] and was reported as nonreactive [<0.5 COI (cutoff interval)], borderline (0.5 to <1.0 COI), or reactive (≥1.0 COI]. In this study, CMV infection was used as a binary variable, with the borderline and nonreactive CMV groups combined. Height was reported in inches, and weight was reported in pounds. Height and weight data were obtained for half of all participants in 2014 and for the other half in 2016. BMI was calculated as weight in lbs/ (height in inches)$^2$ *703. We used grip strength to assess frailty because grip strength is a strong indicator of frailty[72,73]. Grip strength was measured using a Smedley spring-type hand dynamometer. Two measurements were taken on each hand. The maximum grip strength of the dominant hand was used in this study. Grip strength was measured for half of the study population in 2016 and the other half in 2014.

**Immune cells.** Peripheral blood mononuclear cells (PBMCs), isolated from whole blood, were used to measure blood cell types using flow cytometry[74]. All measurements were performed either on a LSRII flow cytometer or a Fortessa X20 instrument (BD Biosciences). Immunophenotyping data were analyzed using OpenCyto and FlowAnnotator, as described in previous studies[42,75].

### Statistical Analysis

The analyses described below were conducted using SAS version 9.4. We used survey weights in all analyses to account for the complex survey design used in the HRS study[38]. We used the weights specific for the VBS data for Sample A and the weights specific for the DNA methylation data for Sample B (Sample A and Sample B are described below).

The analyses were conducted in two samples: Sample A included participants who reported their SA and had biomarker measures used to calculate KDM-BA and PhenoAge, and Sample B included participants with data on ECs. Among 9193 participants attended the 2016 VBS study, Sample A excluded 2424 participants with missing values for any clinical biomarker used to calculate KDM-BA, 414 with missing values for any biomarker used to calculate PhenoAge, 1240 who did not answer questions about SA, as well as 1 participant missing in cancer status, 70 missing death status, 52 missing survey weights, 16 missing race/ethnicity, 215 missing BMI, and 1 missing smoking status. This resulted in 5501 participants, which included 946 cancer survivors and 4555 controls, in Sample A (Supplementary Fig. 1). Among 4018 participants with ECs, Sample B excluded 3 participants of missing in cancer status, 63 missing death status, 143 missing survey weights, 11 missing race/ethnicity, 403 missing BMI, 1 missing smoking status, and 7 missing CMV status. This resulted in 3387 participants, which included 582 cancer survivors and 2805 controls, in Sample B (Supplementary Fig. 2). Since the exclusion of participants with missing information might change the distributions of variables of interest, we compared the distributions of main demographic and lifestyle factors among participants who were included and not included in the study.

Since these aging constructs, except for mPOA ($r = 0.03$ with CA) and Zhang Score ($r = 0.31$ with CA), were moderately or strongly correlated with CA (Supplementary Tables 3 and 4), we estimated age acceleration (abbreviated as Accel) for each aging construct as residuals after regressing the aging construct on CA, allowing us to evaluate the effects of aging construct independent of age. Additionally, in agreement with previous studies, we used the Zhang Score rather than age acceleration for this construct because this is a mortality risk score that was weakly correlated with chronological age[69]. The distributions of aging constructs and characteristics of interest were examined as

weighted mean (standard deviation (SD)) or weighted percentage across cancer status in 2016.

We used weighted logistic regression to estimate weighted odds ratios (ORs) and 95% confidence intervals (CIs) for the associations between each aging construct and cancer prevalence. A participant was considered having prevalent cancer if they reported being diagnosed with cancer at any time before 2016. We applied weighted Cox proportional hazards regression to examine the association between each aging construct and all-cause mortality among cancer survivors (i.e., participants who reported being diagnosed with cancer at any time before 2016) and controls (i.e., those who reported having no history of cancer in 2016). The follow-up for mortality started from the date of blood collection in 2016 and ended on either death or December 31, 2020, whichever occurred first. The proportional hazards assumption, examined by including an interaction term between follow-up time and each of the aging construct, was not violated in any of the models.

To examine KDM-BA, PhenoAge, and SA (Sample A), we ran two models. Model 1 was adjusted for chronological age, sex, and race/ethnicity. Model 2 (fully adjusted model) was additionally adjusted for education, BMI, smoking status, ever drinking, physical activity, comorbidity index, and CMV infection. In both models, we accounted for survey weights. To examine ECs (Sample B), we additionally adjusted, in both models, for the most prevalent immune cells, i.e., percentage of monocytes, neutrophils, and lymphocytes since immune cells were associated with EC in the previous study by Zhang et al. [2024][44]. We included monocytes, neutrophils, and lymphocytes only because we found that adjustment for these immune cells produced results similar to the adjustment for a broader set of immune cells from Zhang's study that examined neutrophils, eosinophils, basophils, monocytes, naïve and memory B cells, naïve and memory CD4+ and CD8+ T cells, natural killer, and T regulatory cells. To ensure that active cancer treatment at the time of blood draw was not a major contributor to our findings, we conducted a sensitivity analysis that excluded cancer survivors whose cancer diagnosis was within two years of the blood collection and examined the associations of aging constructs with cancer prevalence and mortality in cancer survivors.

We also performed four exploratory analyses. In the first exploratory analysis, we reported mean (SD) for aging constructs among cancer survivors stratified by cancer treatment (chemotherapy, surgery, and radiation therapy). In the second exploratory analysis, we investigated the associations between aging constructs (except for KDM-BA) and the prevalence of the most common cancer types (breast, prostate, lung, and colorectal cancers). We did not examine the association using KDM-BA because the data for specific cancer types in HRS is located in the restricted enclave with limited storage space that is insufficient for the KDM-BA computation. Further, we did not examine associations with mortality among individuals with the most common individual cancers due to the small number of cases for each cancer type. In the third exploratory analysis, we investigated whether the association with mortality in cancer survivors was modified by sex, race/ethnicity (among non-Hispanic White and non-Hispanic Black individuals), chemotherapy (Yes/No), surgery (Yes/No), radiation therapy (Yes/No) or time since cancer diagnosis (<2 years, 2 to 5 years, and >5 years). Of note, we restricted the associations stratified by race/ethnicity to non-Hispanic White and non-Hispanic Black participants due to the limited number of deaths in Hispanic participants and participants from other racial/ethnic groups. We conducted the interaction analyses by stratifying by the variable of interest and also by computing the p-value for the interaction term between aging measures and the variable of interest. In the fourth exploratory analysis, we investigated the associations with total cancer incidence among controls, i.e., cancer-free participants in 2016, by applying weighted Cox proportional hazards regression. The follow-up for incident cancer started from the date of blood collection in 2016 and ended on the date of cancer diagnosis, death, or December 31, 2020, whichever occurred first.

## Reporting summary
Further information on research design is available in the Nature Portfolio Reporting Summary linked to this article.

## Data availability
All HRS data are publicly available on the HRS website (https://hrs.isr.umich.edu/data-products/).

## Code availability
The code used for this analysis can be accessed at: https://github.com/wang8310/Aging_Cancer_HRS.

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

## Acknowledgements

This study was supported by the National Center for Advancing Translational Sciences, grant 1UM1TR004405 (S.W.), the National Cancer Institute, grant R01CA267977 (A.P.), and the National Institute on Aging, grant R01AG060110. The authors gratefully acknowledge support from the University of Minnesota Life Course Center on the Demography and Economics of Aging (P30AG066613) funded through a grant from the National Institute on Aging. The Health and Retirement Study is supported by the National Institute on Aging, grant U01AG009740, and is conducted by the University of Michigan.

## Author contributions

S.W. and A.P. contributed equally to this study. S.W. and P.M. conducted data extraction and formal analysis. S.W., A.P., and P.M. wrote the first draft of the manuscript. S.W. and A.P. wrote the final version of the manuscript. S.W., A.P., P.M., S.V., W.G., A.H.B., H.H.N., and B.T. reviewed the draft of the manuscript, made critical revisions, and approved the final manuscript. A.P. and B.T. supervised the study.

## Competing interests

The authors declare no competing interests.
