## [Transparent Peer Review file · Nature Communications]

Aging Measures and Cancer in the Health and Retirement Study (HRS)

Corresponding Author: Dr Anna Prizment

Version 0:

Reviewer comments:

Reviewer #1

(Remarks to the Author)

This interesting study demonstrated that increased age acceleration in the Horvath, Hannum, Levine, and GrimAge epigenetic clocks, as well as subjective age (SA), was significantly correlated with higher cancer prevalence in both cancer survivors and controls. Among cancer survivors, age acceleration in PhenoAge, Hannum, Horvath, Levine, and GrimAge was strongly associated with increased all-cause mortality, with GrimAge presenting the highest hazard ratio. In cancer-free individuals, mortality was notably associated with POA, KDM-BA, PhenoAge, Hannum, and GrimAge, with GrimAge showing the most substantial association. However, the study did not find an association between aging clocks and cancer risk.

This work is significant because it leverages a large, nationally representative sample from the Health and Retirement Study to provide robust evidence on the relationship between biological age and the risk of mortality in cancer survivors versus cancer-free individuals. This comprehensive approach builds on the existing literature that has identified multiple biomarkers of aging, uniquely highlighting the differential impact of specific epigenetic clocks, especially GrimAge, on health outcomes. In comparison to the established literature, this study offers a more detailed analysis by including multiple aging constructs and their associations with both cancer prevalence and mortality, something that previous studies have often examined in isolation. Additionally, the authors report no association with cancer risk. The findings are particularly valuable as they reinforce the utility of epigenetic clocks, especially GrimAge, as potent indicators of biological age. Therefore, the study's unique contribution lies in its comparative analysis across multiple biological aging constructs and its dual focus on both cancer survivors and the general population.

Here are some suggestions for improvement:

1. The authors suggest that age acceleration might be linked to the effects of cancer treatments, yet this potential influence was not included in the analysis. Was information on the history of cancer treatment available in HRS?

2. What are the psychiatric problems that the authors evaluated? Other variables that possibly would interact with BA are physical activity, diet, and lifestyle factors (smoking and alcohol). I believe HRS has variables on physical activity, education, and alcohol.

3. The work supports the conclusions and claims but would benefit from the analysis of additional variables and details. Details regarding interactions are missing; the authors report interaction results, such as "gender modified the association between GrimAgeAccel." Were other variables evaluated? It would benefit readers to mention details in the methods section. Participants were excluded if characteristics of interest or cancer-related characteristics were missing. The authors performed a complete case analysis; it would be beneficial to provide the excluded missing data to ensure it has not resulted in biased results. Is the cause of death available in the dataset?

4. Adding details on missing data, interactions, and cancer characteristics would make the work reproducible.

5. Given the focus on biological aging, it is unclear why aging phenotypes are not characterized. Is it possible to characterize the burden of chronic comorbidities and functional status (frailty) in this study population?

Minor comments:

6. Reference 10 calls Pace of Aging (POA) as DunedinPoAm. Why not refer to it the same way?

Reviewer #2

(Remarks to the Author)

This article reports analysis of associations between cancer prevalence and several epigenetic clocks in data from about 4000 older adults in the US Health and Retirement Study. They also conduct analysis of two blood-chemistry clocks in ~10k individuals. They find that the clocks most strongly associated with healthspan tend to indicate older biological age/ faster biological aging in individuals with cancer vs. those who do not report having cancer. No associations were observed between clocks measured in 2016 and participant reports of new cancer diagnoses through 2020.

This article covers ground already well travelled by cancer cohort studies, e.g., Dugue et al. 2021 JNCI, Dugue et al. 2018 J Cancer, Guida et al. Nat Cancer 2024 and causal analysis via mendelian randomization e.g. Berstein et al. 2022 eLife. It's not clear what the value add is here given the substantial literature on this topic from datasets with more rigorous and specific measurement of cancer. (The HRS itself is population representative. But the DNAm sample is not.)

Some day, when follow-up from biological age measurement is more extensive, the prospective analysis of biological age/pace of aging and cancer incidence will be of interest in this cohort. But for now, this simple analysis of participant-reported cancer outcomes from a single dataset doesn't seem to generate an advance commensurate with what I associate as the standard for a general science journal like Nat Comms.

Reviewer #3

(Remarks to the Author)

The authors of the manuscript presented the results of an analysis of epigenetic clocks and their association with cancer prevalence and mortality in a cohort of the Health and Retirement Study (HRS) with a total of 582 cancer survivors and 2,805 controls. Cancer prevalence was found to be associated with age acceleration as measured by the Hannum, Horvath, Levine, and GrimAge clocks and subjective age. Age acceleration was also associated with mortality among cancer survivors. Although the study is interesting and provides valuable and novel data, the main limitation of the study is the lack of a broader discussion and possible analyses that could indicate the causality of the association between epigenetic clocks and cancer prevalence, whether biological aging is a cause of cancer or rather a consequence. This could be explored using SNP data, if available, and a Mendelian randomization approach. This type of analysis would greatly enhance the value of the study. Furthermore, the discussion is full of dry facts and lacks groundbreaking ideas. For example, the authors should point out that an association was observed for different generations of epigenetic clocks, including first-generation clocks, which often have lower predictive power. Major and minor comments are listed below.

- 1) A great deal of value could be added by analyzing another epigenetic clock. This is the Mortality Risk Score by Zhang et al. [Nat Commun 2017 Mar 17:8:14617. doi: 10.1038/ncomms14617.] particularly since the authors analysed mortality in their sample set.
- 2) The concept of model 2 appears in the results before any explanation or mention of model 1. Since the Materials and Methods are at the end of the paper, the description of the results must be clear to the recipient in the Results section. Therefore, explanations of models 1 and 2 should be provided.
- 3) No correction for blood cell count was applied in models 1 and 2, which could significantly bias the results.
- 4) At a minimum, chronological age, sex and blood cell count should be included as covariates in model 1.
- 5) Table 2 does not include an assessment of significance (p). Can this be added by the authors?
- 6) Line 120 in Results "GrimAge exhibited the highest OR with lung cancer prevalence..." – please provide p value
- 7) It is not clear whether a model 2 stands for the same thing as a fully adjusted model? This should be consistent throughout the manuscript.
- 8) Line 147 - It is not clear the goal and approach/methods applied for the sensitivity analysis
- 9) I suggest changing the name of the POA clock to the more commonly used mPoA. Also, it's not clear why the authors didn't use the updated version of the clock, i.e. PACE
- 10) Methods. Although authors stated that DNA methylation methods are described in another study, some basic information should be provided, including the type of the array, was it the same for all individuals etc.
- 11) There is too little information about the method used to calculate the epigenetic aging parameters and the EAA parameter. Was the epigenetic age determined using Horvath's online tool or a specific R script (differences may be visible to some extent)
- 12) Authors excluded minor skin cancer from the study, and rationale behind should be better explained
- 13) Line 347 - There is some inaccuracy; in the description of the methods, the authors state that Model 1 was only adjusted for chronological age, and in Table 2 there is a description that the adjustment was for chronological age and survey weights. It is also unclear what the authors mean by survey weights.

Version 1:

Reviewer comments:

Reviewer #2

(Remarks to the Author)

The authors have not addressed my comments. As noted, the same research question has been addressed with more comprehensiveness and rigor in other datasets. The fact that this particular set of nine clocks may not have been analyzed together previously is a thin basis for publication. Moreover, I don't believe the survey weights HRS estimated for their DNAm sample don't allow the results of this analysis to be generalized to the US population. If the authors feel that these weights are doing the important work of making their sample representative of the US, they should demonstrate the success by comparing rates of cancer estimated from their weighted sample to population data for the US from SEER or another source.

Reviewer #3

(Remarks to the Author)

The authors responded to all comments and added to the descriptions and suggested analyses. I have no further comments on the manuscript.

Reviewer #4

(Remarks to the Author)

The authors have demonstrated responsiveness to the previous review, and the manuscript is much improved. I do, however, have one additional suggestion:

1. Please present the complete Model 1 and Model 2 in Tables 2 and 3. That is, add the OR/HR and 95% CI for each of adjusted variables. It would be helpful for the reader to see the magnitudes of the effects of the aging constructs/biomarkers in relation to the effects of the known risk factors of age, sex, race, education, BMI, smoking, drinking, physical activity and comorbidity indices.

Version 2:

Reviewer comments:

Reviewer #3

(Remarks to the Author)

I accept the authors' responses and the changes made.

We appreciate the reviewers' comments. We have responded to each comment in a point-by-point response and believe that the reviewer feedback has substantially strengthened the manuscript.

REVIEWER COMMENTS

Reviewer #1 (Remarks to the Author): expertise in epidemiology and aging

1. The authors suggest that age acceleration might be linked to the effects of cancer treatments, yet this potential influence was not included in the analysis. Was information on the history of cancer treatment available in HRS?

Response: We have now included the information about cancer treatment which was available by self-report, in the revised manuscript. Cancer treatment information was collected (from 1992 to 2016) in response to the question "Since [previous interview], what sort of treatments have you received for cancer?" Participant responses were categorized into eight categories: chemotherapy, surgery, radiation, medications/treatments for symptoms, biopsy, x-ray, other, or none.

In response to the reviewer's comment, we have now reported the percentages of cancer survivors who had chemotherapy, radiation, and surgery in Table 1. We have tested whether age acceleration differed by treatment status. Here is what we included in the Results section: *"We found that, among cancer survivors, those treated with chemotherapy (vs. those without) had, on average, higher age acceleration for PhenoAge, subjective age (SA), Horvath, Levine, and GrimAge; however, these differences did not reach statistical significance (Supplemental Table 5). In addition, the mean age acceleration for Levine and GrimAge were higher among those who received radiation therapy compared to those without, while the mean age acceleration for Horvath was higher among those who had surgery compared to those without surgery, but none of these differences reached statistical significance (Supplemental Table 5). Although these non-significant results may reflect the true absence of differences by treatment status, the similar trends across different treatments and different aging measures suggest that the absence of significant differences may be explained by a limited sample size in these subgroup analyses."*

Further, we conducted an exploratory analysis of aging measures and mortality among cancer survivors stratified by treatment (Supplemental Tables 9-11). Here is what we added in the Results section:

"We found that, compared to cancer survivors who did not receive chemotherapy, the associations with mortality among those who received chemotherapy was significantly stronger for age acceleration for SA (p -interaction=0.015) and appeared to be stronger for age acceleration for PhenoAge with a borderline significance (p -interaction=0.060). Similar trends were observed for age acceleration for Levine and GrimAge as well as Zhang Score but these p -values for interactions with chemotherapy did not reach statistical significance; however, the number of deaths among cancer survivors who self-reported information about chemotherapy was limited. Although there were no statistically significant interactions of aging constructs with surgery (Yes/No) or radiation therapy (Yes/No) in relation to mortality among cancer survivors, there was a pattern of stronger associations with mortality observed among cancer survivors who had surgery (or radiation) vs. those who did not have surgery (or radiation) for several aging constructs."

Hence, we concluded that several aging metrics, especially PhenoAge, a clock based on clinical biomarkers, and second-generation epigenetic clocks (GrimAge and Levine) may differentiate not only between cancer survivors and cancer-free persons but also between those with chemotherapy vs. no chemotherapy and radiation vs. no radiation. Future larger studies are warranted that will investigate how cancer therapies impact biological aging in cancer survivors.

2. What are the psychiatric problems that the authors evaluated? Other variables that possibly would interact with BA are physical activity, diet, and lifestyle factors (smoking and alcohol). I believe HRS has variables on physical activity, education, and alcohol.

Response:

Psychiatric problems were captured with the following self-report question: "Have you ever seen a doctor for psychiatric, emotional or nervous problems?"

The reviewer is correct that HRS collected information about physical activity, education, and alcohol. In the revised manuscript, we have added the description of physical activity (as a weighted sum of score as described in [1]), education (less than high school/high school/greater than high school), and alcohol intake (ever drinker/never drinker) in the methods and included their distribution by cancer status in Table 1. Also, we have now adjusted all analyses in Model 2 (fully adjusted model) for physical activity, education, and alcohol intake in cancer survivors and cancer-free controls. We also tested statistical interactions with additional variables including race/ethnicity and treatment in cancer survivors in addition to sex and time since diagnosis that we did before. We have added in the methods now that we conducted interaction analyses by stratifying by the variable of interest and also by computing p-value for the multiplication term between aging measure and the variable of interest. We did not test interactions with other variables since this analysis would lead to multiple comparisons and potential false positive results due to the limited sample size for interaction analyses.

Diet information in HRS was collected in a very small sample (less than 30% of the whole study participants). Additionally, diet information was collected in 2013, i.e., earlier than the blood collection in 2016. Thus, we did not include the analysis of diet in this manuscript.

Reference:

1. Wu C, Geldhof GJ, Xue QL, Kim DH, Newman AB, Odden MC. Development, Construct Validity, and Predictive Validity of a Continuous Frailty Scale: Results From 2 Large US Cohorts. *Am J Epidemiol.* 2018;187(8):1752-1762. doi:10.1093/aje/kwy041

3. The work supports the conclusions and claims but would benefit from the analysis of additional variables and details. Details regarding interactions are missing; the authors report interaction results, such as "gender modified the association between GrimAgeAccel." Were other variables evaluated? It would benefit readers to mention details in the methods section.

Response: We have added the analysis and details regarding interaction requested by the reviewer, as described in our response to Question 2 above.

4. Participants were excluded if characteristics of interest or cancer-related characteristics were missing. The authors performed a complete case analysis; it would be beneficial to provide the

excluded missing data to ensure it has not resulted in biased results. Is the cause of death available in the dataset?

Response: Unfortunately, cause of death is not available in the HRS study, thus we analyzed all-cause mortality only.

In the initial submission of our manuscript, we described participants' exclusion in Supplemental Figures 1 and 2. Now we've provided a description of the excluded participants in the Methods section:

“Among 9,193 participants attended the 2016 VBS study, Sample A excluded 2,424 participants with missing values for any clinical biomarker used to calculate KDM-BA, 414 with missing values for any biomarker used to calculate PhenoAge, 1240 who did not answer questions about SA, as well as 1 participant missing in cancer status, 70 missing death status, 52 missing survey weights, 16 missing race/ethnicity, 215 missing BMI, and 1 missing smoking status. This result in 5,501 participants, which included 946 cancer survivors and 4,555 controls, in Sample A (Supplemental Figure 1). Among 4,018 participants with ECs, Sample B excluded 3 participants of missing in cancer status, 63 missing death status, 143 missing survey weights, 11 missing race/ethnicity, 403 missing BMI, 1 missing smoking status, and 7 missing CMV status. This resulted in 3,387 participants, which included 582 cancer survivors and 2,805 controls, in Sample B (Supplemental Figure 2).”

Now we've added a table that compared the distributions of main demographic and lifestyle factors among participants who were included and not included (Supplemental Table 2). We found that included participants tended to be non-Hispanic White and have a higher education level in both Sample A and Sample B. In addition, there was a smaller percentage of current smokers among participants included in Sample A. The difference among participants who were included and not included may be explained by the fact that we included only those who answered questions about SA and characteristics of interest, e.g., race/ethnicity, education, smoking status, BMI, etc. Although the populations included and not included were not the same, the inclusion of only the complete sample would make our sample less generalizable to the whole U.S. population but the findings of our study are unlikely to be biased.

Supplementary Table 2. Distributions^a of demographics and lifestyle factors among participants included and not included in the study with the 2016 Venous Blood Study (VBS) data for Sample A and with the DNA methylation data for Sample B, HRS

Sample A			
	Included in the study	Not included	P-value ^b
Mean chronological age, years (SD)	68.02 (0.25)	69.37 (0.37)	<0.0001
Male, %	46	45.7	0.81
Race/Ethnicity, %			
Non-Hispanic White	83.4	69	<0.0001
Non-Hispanic Black	7.4	14.3	
Hispanic	4.3	8.2	
Other	4.9	8.5	
Education level, %			
< High School	10.6	20.8	<0.0001
High School	29.2	29	
Greater than High School	60.2	50.2	
Mean BMI, kg/m ² (SD)	29.8 (0.10)	30.5 (0.19)	0.0006
Smoking status, %			

Current smokers	9.9	12.9	0.003
Former smokers	44.1	43.7	
Sample B			
	Included in the study	Not included	P-value ^b
Mean chronological age, years (SD)	68.5 (0.29)	68.6 (0.63)	0.861
Male, %	45.9	45.6	0.913
Race/Ethnicity, %			
Non-Hispanic White	78.4	72	0.005
Non-Hispanic Black	10.1	9.7	
Hispanic	5.8	8.9	
Other	5.7	9.4	
Education level, %			
< High School	13.3	18.9	<0.0001
High School	29.4	33.1	
Greater than High School	57.3	48	
Mean BMI, kg/m ² (SD)	29.9 (0.13)	29.9 (0.79)	0.679
Smoking status, %			
Current smokers	10.9	11.6	0.934
Former smokers	44.2	43.8	

^aResults were accounted for survey weights.

^bP-values were calculated for chi-square test for categorical variables and t-test for continuous variables.

4. Adding details on missing data, interactions, and cancer characteristics would make the work reproducible.

Response: Thank you for your comment. We have added the requested information to the Methods and Results section as described in our responses to Questions 1-3 and in the revised manuscript. Also, in the revised manuscript we have provided additional details regarding the information collected about cancer characteristics including time since cancer diagnosis, date of cancer diagnosis, type of cancer, and cancer treatment (see our responses to Question 1).

5. Given the focus on biological aging, it is unclear why aging phenotypes are not characterized. Is it possible to characterize the burden of chronic comorbidities and functional status (frailty) in this study population?

Response: We thank the reviewer for a thoughtful suggestion. We have added the following metrics into Table 1:

- Comorbidities: hypertension, lung disease, cardiac disorders, stroke, arthritis, diabetes, and psychiatric problems (these are the chronic conditions we used to calculate a comorbidity index in our study).
- Frailty assessed by grip strength because grip strength is a strong predictor for frailty [1].

Now we reported that, compared to cancer-free participants, cancer survivors were more likely to have hypertension, cardiac disorders, stroke, arthritis, and diabetes, as well as lower grip strength, i.e. tended to be frail.

References:

1. Syddall H, Cooper C, Martin F, Briggs R, Aihie Sayer A. Is grip strength a useful single marker of frailty?. *Age Ageing*. 2003;32(6):650-656. doi:10.1093/ageing/afg111

Minor comments:

6. Reference 10 calls Pace of Aging (POA) as DunedinPoAm. Why not refer to it the same way?

Response: We now use the term “mPOA” (as suggested by Reviewer 3) throughout the manuscript as it is emerging as the standard nomenclature in the published literature. When we introduced this abbreviation, we now clarified that it is also called DunedinPoAm.

Reviewer #2 (Remarks to the Author): expertise in epidemiology of cancer risk

This article covers ground already well travelled by cancer cohort studies, e.g., Dugue et al. 2021 JNCI, Dugue et al. 2018 J Cancer, Guida et al. Nat Cancer 2024 and causal analysis via mendelian randomization e.g. Berstein et al. 2022 eLife. It's not clear what the value add is here given the substantial literature on this topic from datasets with more rigorous and specific measurement of cancer. (The HRS itself is population representative. But the DNAm sample is not.)

Some day, when follow-up from biological age measurement is more extensive, the prospective analysis of biological age/pace of aging and cancer incidence will be of interest in this cohort. But for now, this simple analysis of participant-reported cancer outcomes from a single dataset doesn't seem to generate an advance commensurate with what I associate as the standard for a general science journal like Nat Comms.

Response: This manuscript advances the field of cancer and aging by examining nine well-validated aging measures and their associations with mortality in cancer survivors and those without cancer within the same study. To our knowledge, there have been no studies that compared the performance of different aging measures across adult cancer survivors, while only one recently published study examined seven aging measures in survivors of childhood cancer. Several other previous studies examined either a single aging measure or a couple of measures in adult cancer survivors. An important distinction of this work compared to prior studies is that it examined cancer survivors aged 70 years or older, who represent a quickly growing group with multiple commodities and require integrated age-sensitive cancer care. In contrast, the majority of previous studies were either focused on survivors of childhood cancer or cancer survivors aged younger than 70 years. The risk stratification based on epigenetic age may help predict their health and select cancer treatment. This is important at both clinical and at public health level. An additional novelty stems from adjusting for immune cells that were measured at the same time as aging measures. Immune cells have been shown to be associated with age acceleration for epigenetic clocks in a recent publication [1], but the previous analyses of epigenetic clocks and cancer have not adjusted for immune cells.

The comment that the DNAm sample is not population representative: Although the DNAm assays in HRS were done on a non-random subsample among individuals who participated in the 2016 Venous Blood Study (VBS), HRS developed survey weights specific for this subsample, making it fully representative of the entire HRS sample [3].

References:

1. Zhang, Ze et al. “Deciphering the role of immune cell composition in epigenetic age acceleration: Insights from cell-type deconvolution applied to human blood epigenetic clocks.” *Aging cell* vol. 23,3 (2024): e14071. doi:10.1111/accel.14071
2. Crimmins E, Kim JK, Fisher J, Faul JD. *HRS Epigenetic Clocks – Release 1*. Ann Arbor, MI: Survey Research Center, Institute for Social Research, University of Michigan; 2020.

Reviewer #3 (Remarks to the Author): expertise in epigenetic clock algorithms

The authors of the manuscript presented the results of an analysis of epigenetic clocks and their association with cancer prevalence and mortality in a cohort of the Health and Retirement Study (HRS) with a total of 582 cancer survivors and 2,805 controls. Cancer prevalence was found to be associated with age acceleration as measured by the Hannum, Horvath, Levine, and GrimAge clocks and subjective age. Age acceleration was also associated with mortality among cancer survivors. Although the study is interesting and provides valuable and novel data, the main limitation of the study is the lack of a broader discussion and possible analyses that could indicate the causality of the association between epigenetic clocks and cancer prevalence, whether biological aging is a cause of cancer or rather a consequence. This could be explored using SNP data, if available, and a Mendelian randomization approach. This type of analysis would greatly enhance the value of the study. Furthermore, the discussion is full of dry facts and lacks groundbreaking ideas. For example, the authors should point out that an association was observed for different generations of epigenetic clocks, including first-generation clocks, which often have lower predictive power. Major and minor comments are listed below.

Response: We agree with the reviewer that our discussion would be benefitted from more depth. To address this comment, we have expanded our discussion in the revised manuscript. Now we provided a more structured summary of the results, including conclusions about the performance of epigenetic clocks from different generations in cancer survivors and cancer-free individuals and the performance of aging measures across cancer treatments, and added our thoughts on aging in cancer survivors. The main points that we added are presented below.

1. In the Discussion, we underscored the importance of studying aging measures in older cancer survivors in the first paragraph and in the paragraph on the “Strengths and limitations” of the Discussion:

“BA measures the accumulated life course changes in biological systems, which differ between individuals of the same chronological age (CA), and therefore may provide information about the aging process in addition to the information provided by CA. Additionally, BA may be modified by anti-aging lifestyle interventions or drugs that are currently under active development. Importantly, because aging process and age-related diseases share the same basic molecular mechanisms, intervening on BA is expected to slow down the development of multiple age-related diseases simultaneously. This underscores the importance of identifying the optimal BA in older populations and particularly in cancer survivors, a group of vulnerable individuals that have increased morbidity and mortality compared to cancer-free individuals.”

“As already discussed, the most important strength of our study is the comparisons of multiple aging measures in cancer survivors and cancer-free individuals within the same study. Importantly, our study included elderly cancer survivors aged 70 years or older, which represent a quickly growing group that needs cancer care tailored for this population.”

2. We agree with the reviewer that the Mendelian approach would help establish causality. Unfortunately, while we have requested the HRS genetic data through NIAGADS, we anticipate

it will take several more months to obtain access to this data from HRS. Thus, we are not proposing this in the manuscript. While causality cannot be examined with prevalent cancer, the comparisons of the prospective associations with mortality in cancer survivors and cancer-free individuals answered the reviewer's question about whether cancer is a cause of the accelerated aging or whether accelerated aging is the cause of cancer. Here is what we added to the discussion:

“Compared to cancer survivors, in cancer-free individuals, there was a trend of weaker associations with mortality for several epigenetic clocks (Hannum, Levine, and GrimAge). For instance, in cancer survivors but not in cancer-free individuals, there were significant associations with mortality for age acceleration for Hannum, Levine, and GrimAge. These findings support the conclusion from a recent NHANES study, which found a pattern of a stronger association between PhenoAgeAccel and mortality in cancer survivors than controls. Most likely, specific features in cancer survivors, such as inflammatory processes associated with cancer itself and cancer treatments, may contribute to the aging process in cancer survivors.”

Our newly added analyses on cancer treatment also helped answer the reviewer's question about whether cancer is a cause of the accelerated aging or whether accelerated aging is the cause of cancer. In the revised manuscript, we have now examined treatment types (chemotherapy, radiation, and surgery) which allowed us to study the levels of age acceleration by treatment types and study the associations between age acceleration and mortality stratified by treatment. Here is what we added in the discussion:

“Cancer survivors may experience accelerated aging due to both cancer itself and cytotoxic cancer treatments, particularly chemotherapy and radiation therapy, which may cause epigenetic changes and cellular senescence features such as telomere shortening and alterations in DNA repair genes. In turn, these changes individually or in combination contribute to accelerated aging phenotypes in cancer survivors. In agreement with the hypothesized mechanisms, in our study, age acceleration for several aging constructs (PhenoAge, SA, Horvath, Levine and GrimAge) tended to be higher in cancer survivors who received treatment (chemotherapy, radiation, or surgery) compared to those who did not; however, all these differences did not reach statistical significance. In addition, in the analysis of mortality stratified by cancer treatment, we found a pattern of stronger associations with mortality for several aging constructs among cancer survivors with treatment vs. those without. Although not all results were significant, the similar trends observed for different treatments and different aging measures suggest that the absence of significant differences may be explained by a limited sample size in these subgroup analyses. An additional limitation of our analysis stratified by treatment is that cancer treatment was self-reported. Future studies with more accurate cancer treatment data are warranted to compare the aging process among cancer survivors with and without treatment and thus could help improve cancer treatment protocols, which are important at both clinical and public health level.”

3. Per reviewer's suggestions, we expanded on the comparison of aging measures that we examined and their relevant utility in cancer survivors and controls. Here is what we included in the Discussion section:

“Furthermore, our findings suggest that both first-generation (Hannum) and second-generation (Levine, GrimAge, Zhang Score) ECs are associated with mortality in cancer survivors. However, in our study, second-generation ECs (i.e., GrimAge, Levine, and Zhang Score) showed stronger associations with mortality in cancer survivors compared to first-generation

ECs (Hannum and Horvath). This may be explained by the fact that second-generation ECs were trained on mortality, while first-generation ECs were trained on chronological age. Thus, it was expected that second-generation ECs were more strongly associated with mortality [3].”

“Moreover, in our study, only two aging measures (PhenoAge and ZhangScore) were associated with mortality in both cancer survivors and controls. Although the associations with mortality for PhenoAgeAccel in cancer survivors and controls were slightly weaker compared to those for Zhang Score, the relative inexpensiveness of PhenoAge measurement and easier implementation makes it a promising option for future clinical applications.”

References:

1. Guida JL, Agurs-Collins T, Ahles TA, et al. Strategies to Prevent or Remediate Cancer and Treatment-Related Aging. *J Natl Cancer Inst.* 2021;113(2):112-122. doi:10.1093/jnci/djaa060
2. Wang S, Prizment A, Thyagarajan B, Blaes A. Cancer Treatment-Induced Accelerated Aging in Cancer Survivors: Biology and Assessment. *Cancers (Basel).* 2021;13(3):427. Published 2021 Jan 23. doi:10.3390/cancers13030427
3. Faul JD, Kim JK, Levine ME, Thyagarajan B, Weir DR, Crimmins EM. Epigenetic-based age acceleration in a representative sample of older Americans: Associations with aging-related morbidity and mortality. *Proc Natl Acad Sci U S A.* 2023;120(9):e2215840120. doi:10.1073/pnas.2215840120

1) A great deal of value could be added by analyzing another epigenetic clock. This is the Mortality Risk Score by Zhang et al. [*Nat Commun* 2017 Mar 17:8:14617. doi: 10.1038/ncomms14617.] particularly since the authors analysed mortality in their sample set.

Response: We have now included Zhang [2017] mortality risk score – a second-generation epigenetic clock. We found that Zhang Score was significantly associated with cancer prevalence (Table 2). In addition, Zhang Score was associated with mortality in both cancer survivors and cancer-free controls in the fully adjusted model (including adjustments for demographics, lifestyle factors, medical histories, and immune cells) (Table 3), while age acceleration for several epigenetic clocks from different generations (Hannum, Levine, and GrimAge) was associated with mortality in cancer survivors but not in controls. Moreover, we found a trend of a stronger association with mortality among cancer survivors who had chemotherapy compared to those without chemotherapy for the Zhang Score, but the difference did not reach statistical significance (Supplemental Table 9).

We concluded that “our results suggest the different aging constructs have different associations with cancer prevalence as well as mortality in cancer survivors and cancer-free individuals after adjustment for potential risk factors. Notably, several aging constructs were more strongly associated with mortality in cancer survivors than controls. We conclude that PhenoAge (clinical-marker based aging measure) and ECs from different generations (Hannum, Levine, Zhang score, and GrimAge) hold promise as potential measure of biological age in cancer survivors.”

2) The concept of model 2 appears in the results before any explanation or mention of model 1. Since the Materials and Methods are at the end of the paper, the description of the results must be clear to the recipient in the Results section. Therefore, explanations of models 1 and 2 should be provided.

Response: We've now provided explanations of models 1 and 2 in the Results section.

3) No correction for blood cell count was applied in models 1 and 2, which could significantly bias the results.

Response: We have now adjusted for the most prevalent immune cells: neutrophil percentage, lymphocyte percentage, and monocyte percentage in both models 1 and 2 in the analysis of epigenetic clocks.

In the results section, we have now described why we adjusted for the three most prevalent immune cells: *“We adjusted, in the analysis of ECs, for immune cells because immune cells have been shown to be associated with age acceleration for ECs in a recent publication: Zhang [2024] [1] showed that age acceleration for ECs was associated with immune cells, such as neutrophils, eosinophils, basophils, monocytes, naïve and memory B cells, naïve and memory CD4 + and CD8 + T cells, natural killer, and T regulatory cells. In addition, a previous study of ECs in HRS included the percentage for immune cells, such as CD8 naïve cell, CD4 T cell, CD8 T cell, monocyte, B cells, and natural killer cells, as confounders [2]. We found that the adjustments for these two groups of immune cells produced results similar to the adjustments for monocyte, neutrophil, and lymphocyte. Thus, we decided to adjust for the three most prevalent immune cells.”*

In the methods section we described how immune cells were measured. *“In HRS, peripheral blood mononuclear cells (PBMCs), isolated from whole blood, were used to measure blood cell types using flow cytometry [1]. All measurements were performed either on a LSRII flow cytometer or a Fortessa X20 instrument (BD Biosciences). Immunophenotyping data were analyzed using OpenCyto and FlowAnnotator, as described in previous studies [3, 4]. “*

References:

1. Zhang Z, Reynolds SR, Stolrow HG, Chen JQ, Christensen BC, Salas LA. Deciphering the role of immune cell composition in epigenetic age acceleration: Insights from cell-type deconvolution applied to human blood epigenetic clocks. *Aging Cell*. 2024;23(3):e14071. doi:10.1111/accel.14071
2. Faul JD, Kim JK, Levine ME, Thyagarajan B, Weir DR, Crimmins EM. Epigenetic-based age acceleration in a representative sample of older Americans: Associations with aging-related morbidity and mortality. *Proc Natl Acad Sci U S A*. 2023;120(9):e2215840120. doi:10.1073/pnas.2215840120
3. Barcelo, H., Faul, J., Crimmins, E. & Thyagarajan, B. A Practical Cryopreservation and Staining Protocol for Immunophenotyping in Population Studies. *Curr Protoc Cytom* 84, e35 (2018). <https://doi.org/10.1002/cpcy.35>
4. Hunter-Schlichting, D. et al. Validation of a hybrid approach to standardize immunophenotyping analysis in large population studies: The Health and Retirement Study. *Sci Rep* **10**, 8759 (2020). <https://doi.org/10.1038/s41598-020-65016-x>

4) At a minimum, chronological age, sex and blood cell count should be included as covariates in model 1.

Response: Now, per reviewer' suggestion, we have included chronological age, sex and blood cell count (neutrophil percentage, lymphocyte percentage, and monocyte percentage).

5) Table 2 does not include an assessment of significance (p). Can this be added by the authors?

Response: We have now provided p-value for each association in Table 2.

6) Line 120 in Results “GrimAge exhibited the highest OR with lung cancer prevalence...” – please provide p value

Response: We have now provided the p-value.

7) It is not clear whether a model 2 stands for the same thing as a fully adjusted model? This should be consistent throughout the manuscript.

Response: We agree that it was not clear in the previous version and now have explained that the model 2 and a fully adjusted model are the same and called it Model 2 consistently throughout the manuscript.

8) Line 147 - It is not clear the goal and approach/methods applied for the sensitivity analysis

Response: Now we have explained the goal and approach/methods for the sensitivity analysis in text: *“To ensure that active cancer treatment at the time of blood draw was not a major contributor to our findings for the association of aging constructs with cancer prevalence and mortality among cancer survivors, we conducted a sensitivity analysis. In this sensitivity analysis, we excluded participants who developed cancer within two years of blood collection – the time window where most cancer patients complete their treatment, and reran the analyses for cancer prevalence and mortality among cancer survivors after this exclusion.”*

9) I suggest changing the name of the POA clock to the more commonly used mPoA. Also, it's not clear why the authors didn't use the updated version of the clock, i.e. PACE

Response: We changed the name of the clock to mPOA. The data on the updated version of this clock (i.e. PACE) has not been publicly released by HRS.

10) Methods. Although authors stated that DNA methylation methods are described in another study, some basic information should be provided, including the type of the array, was it the same for all individuals etc

Response: Now, in the Methods section, we have added that *“DNAm was measured from DNA extracted from the buffy coat using the Infinium Methylation EPIC BeadChip by the University of Minnesota Genomics Center. DNA samples were randomized across analytic plates by age, cohort, sex, education, and race/ethnicity along with 39 pairs of blinded duplicates. Analysis of duplicate samples showed a correlation >0.97 across all CpG sites. Data preprocessing and quality control were performed using the minfi package in R. A total of 3.4% of the methylation probes were removed from the final data because their detection P-value fell below the threshold of 0.01 (n = 29,431 out of 866,091). The same DNAm measurements were used for all the individuals and high-quality data was available for 4,018 individuals [1].”*

Reference:

1. Crimmins E, Kim JK, Fisher J, Faul JD. HRS Epigenetic Clocks – Release 1. Ann Arbor, MI: Survey Research Center, Institute for Social Research, University of Michigan; 2020.

11) There is too little information about the method used to calculate the epigenetic aging parameters and the EAA parameter. Was the epigenetic age determined using Horvath's online tool or a specific R script (differences may be visible to some extent)

Response: In the original manuscript submission, we provided information in Supplemental Table 1 on how the epigenetic clocks were originally constructed. Now we added more information on each clock in that table and made more clear in the Methods section that this information can be found in Supplemental table 1. Also we added in the main text that epigenetic clocks in HRS were calculated using R scripts [1, 2]. We also underscored in the Methods and Results sections that age acceleration for each clock (except for mPOA and Zhang Score) was calculated as residuals after regressing aging clock on chronological age to make the aging measure independent of chronological age. We did not calculate age acceleration for mPOA because it reflects the pace of the aging process and it was not correlated with CA ($r=0.03$), or Zhang Score, because it is a mortality risk score that was only weakly correlated with CA ($r=0.31$) (Supplemental Table 4).

References:

1. Crimmins E, Kim JK, Fisher J, Faul JD. HRS Epigenetic Clocks – Release 1. Ann Arbor, MI: Survey Research Center, Institute for Social Research, University of Michigan; 2020.
2. Ware EB, Higgins Tejera C, Wang H, Harris S, Fisher JD, Bakulski KM. Interplay of education and DNA methylation age on cognitive impairment: insights from the Health and Retirement Study. *Geroscience*. Published online September 26, 2024. doi:10.1007/s11357-024-01356-0

12) Authors excluded minor skin cancer from the study, and rationale behind should be better explained

Response: Non-melanoma skin cancer was not included for two inter-related reasons. First, survival for patients for most non-melanoma skin cancers is excellent (the 5-year relative survival is 99% for basal cell carcinoma (BCC) and slightly less than 95% for squamous cell carcinoma (SCC)). So they are very different from other cancer types. Second, non-melanoma skin cancer cases are usually diagnosed and treated easily in physician's office. It is difficult to keep track of these cancers, and individuals may not be even aware that they have these cancers. In summary, similar to other population-based cohorts, we did not include non-melanoma skin cancer cases in our analyses (similar to other population-based studies on cancer) because they have a very low mortality rate and the self-reported information on their diagnosis is highly unreliable.

13) Line 347 - There is some inaccuracy; in the description of the methods, the authors state that Model 1 was only adjusted for chronological age, and in Table 2 there is a description that the adjustment was for chronological age and survey weights. It is also unclear what the authors mean by survey weights.

Response: We now added explanation in the Results and Methods sections. *"We used Survey to account for the complex survey design used in the Health and Retirement Study [1]. We used the weights specific for the 2016 VBS study for Sample A and the weights specific for the DNA methylation sample for Sample B [1]."*

Reference:

1. Sonnega A, Faul JD, Ofstedal MB, Langa KM, Phillips JW, Weir DR. Cohort Profile: the Health and Retirement Study (HRS). *Int J Epidemiol*. 2014;43(2):576-585. doi:10.1093/ije/dyu067

We appreciate the reviewers' comments. We have responded to each comment in a point-by-point response and believe that the reviewer feedback has substantially strengthened the manuscript.

REVIEWER COMMENTS

Reviewer #2 (Remarks to the Author):

The authors have not addressed my comments. As noted, the same research question has been addressed with more comprehensiveness and rigor in other datasets. The fact that this particular set of nine clocks may not have been analyzed together previously is a thin basis for publication. Moreover, I don't believe the survey weights HRS estimated for their DNAm sample don't allow the results of this analysis to be generalized to the US population. If the authors feel that these weights are doing the important work of making their sample representative of the US, they should demonstrate the success by comparing rates of cancer estimated from their weighted sample to population data for the US from SEER or another source.

Response:

Thank you for an opportunity to respond to your thoughtful comments that have made our paper better. As far as we understand the reviewer has two interconnected concerns: (1) the question on aging in cancer has been examined in other studies, and thus identifying the best aging construct in cancer does not justify our study and (2) the DNA methylation (DNAm) sample in HRS does not reflect the whole HRS cohort and does not allow the results to be generalized to the US population. Please see below our answers to these questions.

(1) We agree with the reviewer that the association between epigenetic clocks and **cancer risk** has been examined in other studies. However, the focus of our paper is on aging in older **cancer survivors**, and fills important gaps in knowledge. The number of cancer survivors is rapidly increasing (cancer survivors aged 60 years or older increased from 8.4 million in 2008 to 14.1 million in 2022),^{1,2} and they have multiple comorbidities in addition to their cancer which are likely to impact their aging process. The declining health of cancer survivors is an unprecedented challenge to the health care system (see reviews^{3,4} for a detailed discussion). Hence, investigating biological aging in older cancer survivors, especially as this phenotype may be modifiable by senomorphics and senolytics, becomes a critical question. It is notable that only a few studies have examined this question,⁵⁻¹⁰ and most of them focused on breast cancer.⁷⁻⁹ A study by Zhang et al. in NHANES [JAMA, 2022]¹⁰ examined the association between phenotypic age (PhenoAge) and mortality in all cancer survivors and individuals without cancer and found a stronger association in those with vs. without cancer. However, that study was not able to compare biological aging across specific cancer types. In addition, it did not have information on methylation or cancer treatment, so it remains unclear which aging metrics work best to capture the aging process in cancer survivors (and whether these metrics are the same or different from cancer-free individuals). Thus, our study provided a comparative analysis across multiple aging constructs, as well as across cancer survivors and cancer-free individuals within the same study. We have underscored that our focus has been on **aging in cancer survivors** during the revision process.

(2) The data in the HRS site [<https://dss.niaqads.org/datasets/ng00153/>], also described in a paper by Jessica Faul et al [PNAS, 2022],¹¹ indicate that the DNAm sample "fully represents the entire HRS sample when weighted." To further address the reviewers comments, we have

compared the cancer prevalence in the HRS DNAm sample with the cancer prevalence in the entire HRS cohort, and with the limited-duration SEER prevalence data (see Table below).¹² The cancer prevalence across different age groups in the HRS DNAm sample and the entire HRS cohort was very similar and also similar to the prevalence reported in SEER. In HRS, we do observe slightly higher cancer prevalence than SEER for the youngest and oldest participants, potentially due to the limited number of HRS participants in those groups. We have also compared the cancer prevalence in HRS DNAm sample with the cancer prevalence in Medicare. For those over 65 years, the cancer prevalence in HRS of 19.8% was similar to the prevalence of 20.1% reported in Medicare Cancer prevalence in 2019,¹³ suggesting that our analysis sample is nationally representative.

Table. Cancer prevalence in 2016 in the HRS DNA methylation (DNAM) sample and the entire HRS cohort, as well as in SEER (limited-duration prevalence) for cancers diagnosed in 1992-2016.			
Age groups	The HRS DNAm sample	The entire HRS cohort	SEER
55-59	9.30%	9.00%	5.78%
60-64	10.50%	11.21%	8.39%
65-69	15.86%	14.52%	12.20%
70-74	20.76%	20.31%	16.71%
75-79	20.64%	22.72%	20.10%
80-84	23.46%	26.53%	22.11%
85+	24.68%	26.66%	21.56%
All age combined	15.74%	15.72%	13.21%

Reviewer #3 (Remarks to the Author):

The authors responded to all comments and added to the descriptions and suggested analyses. I have no further comments on the manuscript.

Reviewer #4 (Remarks to the Author):

The authors have demonstrated responsiveness to the previous review, and the manuscript is much improved. I do, however, have one additional suggestion:

1. Please present the complete Model 1 and Model 2 in Tables 2 and 3. That is, add the OR/HR and 95% CI for each of adjusted variables. It would be helpful for the reader to see the magnitudes of the effects of the aging constructs/biomarkers in relation to the effects of the known risk factors of age, sex, race, education, BMI, smoking, drinking, physical activity and comorbidity indices.

Response: We have now reported the OR/HR and 95% CI for each of adjusted variables in Model 1 and Model 2 in Tables 2 and 3 in Appendix.

References:

1. Parry C, Kent EE, Mariotto AB, Alfano CM, Rowland JH. Cancer survivors: a booming population. *Cancer Epidemiol Biomarkers Prev.* Oct 2011;20(10):1996-2005. doi:10.1158/1055-9965.EPI-11-0729
2. National Cancer Institute. Statistics and Graphs. Retrieved January 31, 2025, from <https://cancercontrol.cancer.gov/ocs/statistics>.
3. Carroll JE, Bower JE, Ganz PA. Cancer-related accelerated ageing and biobehavioural modifiers: a framework for research and clinical care. *Nature reviews Clinical oncology.* Mar 2022;19(3):173-187. doi:10.1038/s41571-021-00580-3
4. Mandelblatt JS, Ahles TA, Lippman ME, et al. Applying a Life Course Biological Age Framework to Improving the Care of Individuals With Adult Cancers: Review and Research Recommendations. *JAMA Oncol.* Nov 1 2021;7(11):1692-1699. doi:10.1001/jamaoncol.2021.1160
5. Gào X, Zhang Y, Boakye D, et al. Whole blood DNA methylation aging markers predict colorectal cancer survival: a prospective cohort study. *Clin Epigenetics.* Nov 30 2020;12(1):184. doi:10.1186/s13148-020-00977-4
6. Dugué PA, Bassett JK, Joo JE, et al. DNA methylation-based biological aging and cancer risk and survival: Pooled analysis of seven prospective studies. *Int J Cancer.* Apr 15 2018;142(8):1611-1619. doi:10.1002/ijc.31189
7. Sehl ME, Carroll JE, Horvath S, Bower JE. The acute effects of adjuvant radiation and chemotherapy on peripheral blood epigenetic age in early stage breast cancer patients. *NPJ breast cancer.* 2020;6(1):1-5.
8. Carroll JE, Crespi CM, Cole S, Ganz PA, Petersen L, Bower JE. Transcriptomic markers of biological aging in breast cancer survivors: a longitudinal study. *J Natl Cancer Inst.* Oct 8 2024;doi:10.1093/jnci/djae201
9. Rentscher KE, Bethea TN, Zhai W, et al. Epigenetic aging in older breast cancer survivors and noncancer controls: preliminary findings from the Thinking and Living with Cancer Study. *Cancer.* Sep 1 2023;129(17):2741-2753. doi:10.1002/cncr.34818
10. Zhang D, Leeuwenburgh C, Zhou D, et al. Analysis of Biological Aging and Risks of All-Cause and Cardiovascular Disease-Specific Death in Cancer Survivors. *JAMA Netw Open.* Jun 1 2022;5(6):e2218183. doi:10.1001/jamanetworkopen.2022.18183
11. Faul JD, Kim JK, Levine ME, Thyagarajan B, Weir DR, Crimmins EM. Epigenetic-based age acceleration in a representative sample of older Americans: Associations with aging-related morbidity and mortality. *Proc Natl Acad Sci U S A.* Feb 28 2023;120(9):e2215840120. doi:10.1073/pnas.2215840120
12. Bluethmann SM, Mariotto AB, Rowland JH. Anticipating the "Silver Tsunami": Prevalence Trajectories and Comorbidity Burden among Older Cancer Survivors in the United States. *Cancer Epidemiol Biomarkers Prev.* Jul 2016;25(7):1029-36. doi:10.1158/1055-9965.epi-16-0133
13. Cancer in Medicare: An American Cancer Society Cancer Action Network Chartbook. https://www.fightcancer.org/sites/default/files/national_documents/acscan-medicare-chartbook.pdf.